# CONFIDENCE DIFFERENCE REFLECTS VARIOUS SUPERVISED SIGNALS IN CONFIDENCE-DIFFERENCE CLASSIFICATION

## ABSTRACT

Training a precise binary classifier with limited supervision in weakly supervised learning scenarios holds considerable research significance in practical settings. Leveraging pairwise unlabeled data with confidence differences has been demonstrated to outperform learning from pointwise unlabeled data. We theoretically analyze the various supervisory signals reflected by confidence differences in confidence difference (ConfDiff) classification and identify challenges arising from noisy signals when confidence differences are small. To address this, we partition the dataset into two subsets with distinct supervisory signals and propose a consistency regularization-based risk estimator to encourage similar outputs for similar instances, mitigating the impact of noisy supervision. We further derive and analyze its estimation error bounds theoretically. Extensive experiments on benchmark and UCI datasets demonstrate the effectiveness of our method. Additionally, to effectively capture the influence of real-world noise on the confidence difference, we artificially perturb the confidence difference distribution and demonstrate the robustness of our method under noisy conditions through comprehensive experiments.

## 1 INTRODUCTION

Weakly supervised learning is an essential research field in machine learning, focusing on training accurate predictive models under conditions of low supervision or imprecise labeling. Due to the difficulty of obtaining precise supervision in real-world scenarios, weakly supervised learning holds significant research value and significance for effectively leveraging limited available supervision information. Consequently, the field of weakly supervised learning has increasingly attracted attention from experts and scholars in recent years, leading to the emergence of many typical weakly supervised learning methods, such as multi-instance learning [32; 30; 24; 19], positive and unlabeled (PU) learning [10; 5; 31; 16; 23], and others.

A prevalent idea in weakly supervised classification involves maximizing the utilization of pointwise weakly supervised information [4], thereby prompting the development of various techniques based on soft labels [18; 26], mixup [28; 21; 27; 9; 7; 13], and others. Nevertheless, it is undeniable that annotating pointwise information in real-world classification problems is a complex and laborious task, further compounded by the personal biases of annotators which frequently exacerbate the probability of inaccuracies. In such scenarios, pairwise comparison information between data points may be more readily obtainable in real-world settings than pointwise information, and it often exhibits greater resistance to biases compared to pointwise semi-supervised information [1]. For instance, in medical diagnosis, accurately determining whether a patient has a disease solely based on their presented symptoms is challenging. However, comparing the symptoms of this patient with those of others provides more accessible information and reduces the probability of misdiagnosis. Extensive research has been conducted on pairwise analysis in numerous binary classification problems, leading to the development of risk minimization functions capable of inducing binary classifiers across various combinations of pairwise similarities, dissimilarities, and unlabeled data [1; 20; 14; 15; 22].

In recent work, pairwise comparison (Pcomp) classification has shown that in tackling difficult point labeling tasks, people can more easily gather comparative information between two examples,

constituting a form of weakly supervised information [4]. However, in real-world application scenarios, individuals may not only distinguish which of two examples is more likely to be classified as positive over the other but also gauge the extent of the disparity in their confidence levels regarding positivity. In light of this framework, Wang et al. introduced a new pairwise weakly supervised classification problem called confidence-difference (ConfDiff) classification, and proposed the corresponding ConfDiff method [22]. To establish confidence difference, the ConfDiff method first utilizes binary-labeled data to train a probability classifier. Subsequently, unlabeled data pairs are fed into the classifier to generate posterior probabilities, from which confidence difference are computed based on the differences between these posterior probabilities. However, through the analysis of the various supervised signals in the ConfDiff method, we identify that ConfDiff method encourages unlabeled data pairs to predict opposite classes from both experimental and theoretical perspectives. This prediction direction is valid when the confidence difference is large. However, when the confidence difference is small, the instances may belong to the same or different classes, and such a predictive tendency may lead to samples from the same class being incorrectly classified as belonging to different classes, thereby introducing noisy supervisory signals.

To handle this problem, in this paper, we concentrate on mitigating the impact of inaccurate predictions when confidence differences are small. Specifically, we analyze the different supervised signals induced by varying confidence differences in the ConfDiff method. We find that pairwise instances with small confidence differences tend to introduce noisy supervised signals, while those with larger confidence differences provide more reliable supervision. Based on this observation, we propose a ConfDiff classification method that incorporates consistency regularization. By partitioning the dataset based on the accuracy of predictive information, we introduce a consistency regularization term for the subset with relatively precise predictions, encouraging the model to produce similar outputs for pairs with small confidence differences. Meanwhile, for the subset with relatively imprecise predictions, we preserve the benefit of reliable supervised signals. Experimental results demonstrate that our method outperforms existing baselines in most cases and exhibits strong robustness even under artificial noise interference.

In summary, this paper's key contributions can be outlined as follows:

- We introduce a method for ConfDiff classification which aims to enhance the accuracy of weakly supervised classification by constructing risk estimator through **C**onsistency **R**isk and **C**onsistency **R**egularization (CRCR).
- We theoretically analyze various supervised signals reflected by different confidence differences in ConfDiff classification. Additionally, we theoretically estimate the error bounds of our proposed method.
- We validate the effectiveness of our method through experiments by comparing it with existing baselines on datasets of varying scales. In addition, the robustness of our method is further validated under the influence of artificially added noise.

## 2 PRELIMINARIES

In this section, we briefly review the problem definitions of binary classification, binary classification with soft labels, and ConfDiff classification.

**Formulation of binary classification**  Binary classification is a typical task in the field of supervised learning, where the goal is to induce a classifier to partition the data space into two categories. Formally, let $\mathcal{X} = \mathbb{R}^d$ and $\mathcal{Y} = \{-1, +1\}$ be the $d$-dimensional feature space and label space, respectively. The dataset $\mathcal{D}_{\text{BC}} = \mathcal{D}_{\text{BC}}^p \cup \mathcal{D}_{\text{BC}}^n$ for binary classification consists of a positive dataset $\mathcal{D}_{\text{BC}}^p$ and a negative dataset $\mathcal{D}_{\text{BC}}^n$:

$$\mathcal{D}_{\text{BC}}^p = \{(\mathbf{x}_i^p \in \mathcal{X}, y_i^p = +1)\}_{i=1}^{n_p}, \ \mathbf{x}_i^p \overset{i.i.d.}{\sim} p(\mathbf{x}|y = +1),$$

$$\mathcal{D}_{\text{BC}}^n = \{(\mathbf{x}_i^n \in \mathcal{X}, y_i^n = -1)\}_{i=1}^{n_n}, \ \mathbf{x}_i^n \overset{i.i.d.}{\sim} p(\mathbf{x}|y = -1),$$

where $n_p$ and $n_n$ denote the number of positive and negative instances, respectively. Let $\pi$ denotes the class prior $p(y = +1)$ and $\ell : \mathbb{R} \times \mathcal{Y} \to \mathbb{R}_+$ denotes a binary loss function. Then binary classification induces a classifier $g : \mathcal{X} \to \mathbb{R}$ from $\mathcal{D}_{\text{BC}}$ by minimizing the following classification risk:

$$R(g) = \pi \mathbb{E}_{p(\mathbf{x}|y=+1)}[\ell(g(\mathbf{x}), +1)] + (1 - \pi)\mathbb{E}_{p(\mathbf{x}|y=-1)}[\ell(g(\mathbf{x}), -1)]. \tag{1}$$

**Formulation of binary classification with soft labels** In binary classification, soft labels typically represent the confidence of each sample belonging to the positive class. Moreover, several studies have shown that using soft labels rather than hard labels can more accurately reflect the data distribution, thus enhancing the accuracy of training binary classifiers. Formally, let $q_i$ denotes the positive confidence of $\mathbf{x}_i$, the dataset $\mathcal{D}_{\text{BC-soft}}$ for binary classification can be defined as follows:

$$\mathcal{D}_{\text{BC-soft}} = \{(\mathbf{x}_i, q_i)\}_{i=1}^n, \ \mathbf{x}_i \overset{i.i.d.}{\sim} p(\mathbf{x}), \ q_i = p(y_i = +1|\mathbf{x}_i),$$

where $p(\mathbf{x}) = \pi p(\mathbf{x}|y = +1) + (1 - \pi)p(\mathbf{x}|y = -1)$. Subsequently, the risk minimization objective function for binary classification with soft labels can be reformulated into the following form:

$$R_{\text{BC-soft}}(g) = \mathbb{E}_{p(\mathbf{x})}[q\ell(g(\mathbf{x}), +1) + (1 - q)\ell(g(\mathbf{x}), -1)]. \tag{2}$$

**Formulation of confidence-difference (ConfDiff) classification** Given that pairwise supervision is typically more accessible than pointwise supervision and it's feasible to not only determine which sample in an unlabeled data pair is more likely positive but also quantify the confidence difference between them in practical scenarios, ConfDiff classification precisely serves as a weakly supervised classification tailored to address this scenario. It specifically deals with weakly supervised classification problems where training data comprises only pairwise unlabeled data and the confidence difference associated with each pair. Formally, let $c_i = c(\mathbf{x}_i, \mathbf{x}_i') = p(y_i' = +1|\mathbf{x}_i') - p(y_i = +1|\mathbf{x}_i)$ be the confidence difference between pairwise unlabeled data $(\mathbf{x}_i, \mathbf{x}_i')$ drawn from a independent identically distribution probability density $p(\mathbf{x}, \mathbf{x}') = p(\mathbf{x})p(\mathbf{x}')$. Considering a pairwise dataset $\mathcal{D}$ drawn from the pairwise unlabeled data and the confidence differences between them:

$$\mathcal{D}_{\text{CD}} = \{((\mathbf{x}_i, \mathbf{x}_i'), c_i)\}_{i=1}^n, \ \mathbf{x}_i \overset{i.i.d.}{\sim} p(\mathbf{x}), \ \mathbf{x}_i' \overset{i.i.d.}{\sim} p(\mathbf{x}).$$

In a recent study, Wang et al. tackled the ConfDiff classification problem in such challenging scenarios [22]. They deduced an unbiased risk estimator for confidence-difference classification from Eq. 1 and trained a binary classifier solely utilizing unlabeled data and confidence differences by minimizing it. The classification risk can be expressed as:

$$R_{\text{CD}}(g) = \frac{1}{2}\mathbb{E}_{p(\mathbf{x}, \mathbf{x}')}[\mathcal{L}(\mathbf{x}, \mathbf{x}') + \mathcal{L}(\mathbf{x}', \mathbf{x})], \tag{3}$$

where $\mathcal{L}(\mathbf{x}, \mathbf{x}') = (\pi - c(\mathbf{x}, \mathbf{x}'))\ell(g(\mathbf{x}), +1) + (1 - \pi - c(\mathbf{x}, \mathbf{x}'))\ell(g(\mathbf{x}'), -1)$. Then Eq. 3 can be refined as follows:

$$R_{\text{CD}}(g) = \frac{1}{2}\mathbb{E}_{p(\mathbf{x}, \mathbf{x}')}[(\pi - c(\mathbf{x}, \mathbf{x}'))\ell(g(\mathbf{x}), +1) + (1 - \pi - c(\mathbf{x}, \mathbf{x}'))\ell(g(\mathbf{x}'), -1)$$
$$+ (\pi + c(\mathbf{x}, \mathbf{x}'))\ell(g(\mathbf{x}'), +1) + (1 - \pi + c(\mathbf{x}, \mathbf{x}'))\ell(g(\mathbf{x}), -1)]. \tag{4}$$

## 3 THE PROPOSED METHOD

In this section, we introduce the proposed noisy ConfDiff method named CRCR.

### 3.1 ANALYSIS OF THE CONFDIFF METHOD

In the ConfDiff method, pairwise instances with confidence differences smaller than 0.5 are prone to introducing noise, while those with larger confidence differences (greater than 0.5) are considered to provide stronger and more reliable supervised signals. To explain this, we consider the general form of many commonly used losses for the prediction function $g(x)$ and target $y$ [29]:

$$\mathcal{L} = \{\ell(g(\mathbf{x}), y)|\ell(g(\mathbf{x}), y) = h(g(\mathbf{x})) - yg(x) \text{ for some function } h\}, \tag{5}$$

Substituting the form of the loss function from Eq.5 into Eq.4, then the classification risk of ConfDiff method can be rewritten as follows and the proof details are presented in the Appendix B:

$$R_{\text{CD}}(g) = \frac{1}{2}\mathbb{E}_{p(\mathbf{x}, \mathbf{x}')}\left[(\frac{1}{2} - c(\mathbf{x}, \mathbf{x}'))\ell(g(\mathbf{x}), +1) + (\frac{1}{2} + c(\mathbf{x}, \mathbf{x}'))\ell(g(\mathbf{x}'), +1)\right]$$
$$+ \frac{1}{2}\mathbb{E}_{p(\mathbf{x}, \mathbf{x}')}\left[(\frac{1}{2} + c(\mathbf{x}, \mathbf{x}'))\ell(g(\mathbf{x}), -1) + (\frac{1}{2} - c(\mathbf{x}, \mathbf{x}'))\ell(g(\mathbf{x}'), -1)\right]$$
$$+ \frac{1}{2}\mathbb{E}_{p(\mathbf{x}, \mathbf{x}')}\left[(1 - 2\pi)(g(\mathbf{x}) + g(\mathbf{x}'))\right]. \tag{6}$$

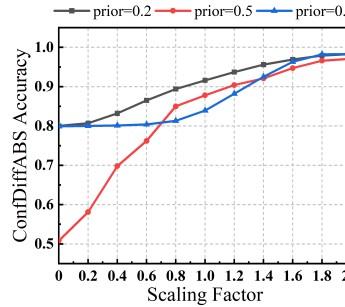 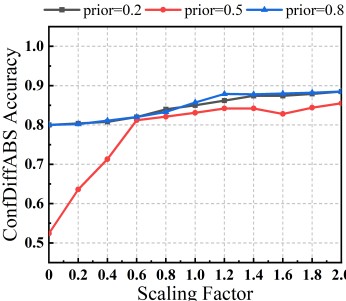

Figure 1: The Accuracy for the binary classifier concerning different proportion of pairwise data with $|c(\mathbf{x}, \mathbf{x}')| > 0.5$ on two benchmark datasets MNIST (left) and CIFAR-10 (right). (The value of the x-axis values $* \min(\pi, 1 - \pi)$ denotes the proportion of pairwise instances with $|c(\mathbf{x}, \mathbf{x}')| > 0.5$.)

where the first and second terms denote the pairwise instance $(\mathbf{x}, \mathbf{x}')$ contrastive losses for positive and negative class predictions, respectively; and the third term serves as a regularization. We first analyze the critical components of the first term, where the weights $\frac{1}{2} - c(\mathbf{x}, \mathbf{x}')$ and $\frac{1}{2} + c(\mathbf{x}, \mathbf{x}')$ determine the contributions of $\mathbf{x}$ and $\mathbf{x}'$ to the positive class prediction loss, respectively. These weights exhibit an inherent balance, as their sum equals 1, indicating that $\frac{1}{2}$ serves as a boundary distinguishing the prediction directions. The weights lie on opposite sides of this boundary, ensuring that one of $\mathbf{x}$ or $\mathbf{x}'$ is encouraged to predict more strongly as the positive class, while the other is encouraged to weaken its positive class tendency (i.e., predict as the negative class). In other words, the first loss term ensures $\mathbf{x}$ and $\mathbf{x}'$ to adjust their predictions in opposite directions, thereby emphasizing the predictive divergence of pairwise instances in the positive class predictions. Similarly, the second loss term forces to diverge in their predictions for the negative class.

Referring to the definition of $c(\mathbf{x}, \mathbf{x}')$, if $|c(\mathbf{x}, \mathbf{x}')| > 0.5$, $\mathbf{x}$ and $\mathbf{x}'$ must belong to different classes; and if $|c(\mathbf{x}, \mathbf{x}')| \leq 0.5$, $\mathbf{x}$ and $\mathbf{x}'$ can belong to the same class or different classes, as the posterior difference is insufficient to surpass the classification threshold. So the prediction trend encouraged by $R_{\mathrm{CD}}$ holds correctly for pairwise instances with $|c(\mathbf{x}, \mathbf{x}')| > 0.5$. However, when $|c(\mathbf{x}, \mathbf{x}')| \leq 0.5$, the prediction trend may lead to samples from the same class being predicted as belonging to different classes, introducing erroneous supervisory signals. Accordingly, we consider that the pairwise instances whose confidence difference are greater than 0.5 contain more supervised signals, but the other ones may result in noisy signals in the existing ConfDiff method.

To further validate this perspective, we conduct experiments on the MNIST and CIFAR-10 by varying the proportion of the pairwise instances with $|c(\mathbf{x}, \mathbf{x}')| > 0.5$. The empirical results (see in Figure 1) illustrate the accuracy of the binary classifier under different proportions of the pairwise instances with $|c(\mathbf{x}, \mathbf{x}')| > 0.5$. We observe a positive correlation between classification accuracy and the proportion value. Notably, when the proportion is 0, the classifier accuracy is approximately 0.5, indicating that the classifier performs nearly at random. These findings demonstrate that the pairwise instances with $|c(\mathbf{x}, \mathbf{x}')| > 0.5$ provide stronger and more reliable supervised signals and dominate the contribution to $R_{\mathrm{CD}}$.

## 3.2 CRCR Method

Based on the discussion in Section 3.1, it is demonstrated that noise signals is introduced when $|c(\mathbf{x}, \mathbf{x}')| \leq 0.5$, while it remains more supervised signals when $|c(\mathbf{x}, \mathbf{x}')| > 0.5$. To address it, we propose setting a threshold $\theta$ to partition the dataset into two subsets: one with relatively precise predictive information (denoted as $D^S$) and the other with comparatively imprecise predictive information (denoted as $D^C$). For $D^C$, we aim to provide additional information to guide the predictions of pairwise instances toward the correct direction. Specifically, for pairwise instances with small confidence differences, we encourage the model to produce more similar outputs for these pairs. To achieve this, we introduce a consistency regularization term that encourages alignment between the confidence difference and the model's outputs. Meanwhile, for $D^S$, we retain the original strategy to preserve the accuracy of predictions driven by this strong guidance. Our objective is to

induce a classifier $g \colon \mathbb{R}^d \to \mathcal{Y}$ from $\mathcal{D}$ by minimizing the expected risk with respect to the data distribution:

$$
\begin{aligned}
R_{\mathrm{CRCR}}(g) = \frac{1}{2}\mathbb{E}_{p_{\mathcal{D}S}(\mathbf{x},\mathbf{x}')}\big[ &\big(\pi - c(\mathbf{x},\mathbf{x}')\big)\ell\big(g(\mathbf{x}),+1\big) + \big(1 - \pi - c(\mathbf{x},\mathbf{x}')\big)\ell\big(g(\mathbf{x}'),-1\big) \\
&+ \big(\pi + c(\mathbf{x},\mathbf{x}')\big)\ell\big(g(\mathbf{x}'),+1\big) + \big(1 - \pi + c(\mathbf{x},\mathbf{x}')\big)\ell\big(g(\mathbf{x}),-1\big)\big] \\
&+ \alpha\mathbb{E}_{p_{\mathcal{D}C}(\mathbf{x},\mathbf{x}')}\big[\big(\frac{1}{\log\left(|c(\mathbf{x},\mathbf{x}')| + \varepsilon\right)}\big) \cdot \|g(\mathbf{x}) - g(\mathbf{x}')\|_2\big],
\end{aligned} \tag{7}
$$

where $\alpha$ denotes the parameter of the consistency regularization term, and $\varepsilon = 1.1$ is a smoothing parameter introduced to mitigate numerical issues when $|c(\mathbf{x},\mathbf{x}')|$ approaches or equals zero. Let $|\mathcal{D}^S| = n_1$ and $|\mathcal{D}^C| = n_2$. Then the risk estimator can be expressed as follows:

$$
\begin{aligned}
\hat{R}_{\mathrm{CRCR}}(g) = \frac{1}{2n_1}\sum_{i=1}^{n_1}\Big(&(\pi - c_i)\ell\big(g(\mathbf{x}_i),+1\big)\Big) + (1 - \pi - c_i)\ell\big(g(\mathbf{x}_i'),-1\big) + (\pi + c_i)\ell\big(g(\mathbf{x}_i'),+1\big) \\
&+ (1 - \pi + c_i)\ell\big(g(\mathbf{x}_i),-1\big)\Big) + \frac{\alpha}{n_2}\sum_{i=1}^{n_2}\left(\frac{1}{\log\left(|c_i| + \varepsilon\right)} \cdot \|(g(\mathbf{x}_i) - g(\mathbf{x}_i')\|_2\right).
\end{aligned} \tag{8}
$$

### 3.3 Analysis of Error Bound

Assuming there exists a constant $C_g$ such that $\sup_{g \in G}\|G\|_\infty \leq C_g$, and another constant $C_\ell$ such that $\sup_{|z|} \leq C_g$ and $\ell(z,y) \leq C_\ell$. Additionally, we presume the binary loss function $\ell(z,y)$ to be Lipschitz continuous with respect to both $z$ and $y$, and to have a Lipschitz constant denoted by $L_\ell$. $\mathfrak{R}_{n_1}(\mathcal{G})$ and $\mathfrak{R}_{n_2}(\mathcal{G})$ denote the Rademacher complexity of unlabeled data $\mathcal{G}$ with size $n_1$ and $n_2$, respectively.

**Theorem 1.** *Let $g^* = \arg\min_{g \in \mathcal{G}}R(g)$ is the minimizer of the true classification risk in Eq.1 and $\hat{g}_{\mathrm{CRCR}} = \arg\min_{g \in \mathcal{G}}\hat{R}_{\mathrm{CRCR}}(g)$ denotes the minimizer of the risk form in Eq.8. Then for any $\delta > 0$, we believe that the following expression holds with a probability at least $1 - \delta$:*

$$
\begin{aligned}
R(\hat{g}_{\mathrm{CRCR}}) - R(g^*) \leq &8L_\ell\mathfrak{R}_{n_1}(\mathcal{G}) + \frac{4\alpha}{\log\left(\varepsilon\right)}\mathfrak{R}_{n_2}(\mathcal{G}) \\
&+ \left(\frac{4C_\ell}{n_1} + \left|\frac{1}{\log\left(\varepsilon\right)} - \frac{1}{\log\left(\theta + \varepsilon\right)}\right|\frac{4\alpha C_g}{n_2}\right)\sqrt{2n\mathrm{In}(2/\delta)}.
\end{aligned} \tag{9}
$$

Due to the space limitation, the proof details are presented in the Appendix A. As $n_1, n_2 \to \infty$, the Rademacher complexities $\mathfrak{R}_{n_1}(\mathcal{G})$ and $\mathfrak{R}_{n_2}(\mathcal{G})$ decrease to zero, and the third term involving $\sqrt{n}/n_1$ and $\sqrt{n}/n_2$ also diminishes. Furthermore, the convergence rates of $\mathfrak{R}_{n_1}(\mathcal{G})$ and $\mathfrak{R}_{n_2}(\mathcal{G})$ are $O(1/\sqrt{n_1})$ and $O(1/\sqrt{n_2})$, while the third term's rate is dominated by $O(\sqrt{n}/n_1)$ and $O(\sqrt{n}/n_2)$. Consequently, as $n \to \infty$, $R(\hat{g}_{\mathrm{CRCR}}) \to R(g^*)$, and the overall convergence rate is characterized by $O\left(\max(\sqrt{n}/n_1, \sqrt{n}/n_2)\right)$.

### 3.4 Empirical Risk Correction

It can potentially lead to severe overfitting problems when the empirical risk becomes negative due to the application of a revised unbiased form. Fortunately, risk correction functions $f(\cdot)$ can be utilized to mitigate this issue. Examples include the absolute value function or the rectified linear unit (ReLU) function. Consequently, the corrected risk estimator can be expressed as follows:

$$
\begin{aligned}
\tilde{R}_{\mathrm{CRCR}}(g) = &\frac{1}{2n_1}f\Big(\sum_{i=1}^{n_1}(\pi - c_i)\ell\big(g(\mathbf{x}_i),+1\big)\Big) + \frac{1}{2n_1}f\Big(\sum_{i=1}^{n_1}(1 - \pi - c_i)\ell\big(g(\mathbf{x}_i'),-1\big)\Big) \\
&+ \frac{1}{2n_1}f\Big(\sum_{i=1}^{n_1}(\pi + c_i)\ell\big(g(\mathbf{x}_i'),+1\big)\Big) + \frac{1}{2n_1}f\Big(\sum_{i=1}^{n_1}(1 - \pi + c_i)\ell\big(g(\mathbf{x}_i),-1\big)\Big) \\
&+ \alpha\frac{1}{n_2}f\left(\sum_{i=1}^{n_2}\left(\frac{1}{\log\left(|c_i| + \varepsilon\right)} \cdot \|(g(\mathbf{x}_i) - g(\mathbf{x}_i')\|_2\right)\right).
\end{aligned} \tag{10}
$$

Table 1: Detailed characteristics of datasets.

| Dataset | #Instance | #Trainset | #Testset | #Fea | Pos Class | Neg Class | Backbone |
|---------|-----------|-----------|----------|------|-----------|-----------|----------|
| MNIST | 70,000 | 15,000 | 5,000 | $28 \times 28$ | 0,2,4,6,8 | 1,3,5,7,9 | 3-layer MLP |
| F-MNIST | 70,000 | 15,000 | 5,000 | $28 \times 28$ | 0,2,4,6,8 | 1,3,5,7,9 | 3-layer MLP |
| K-MNIST | 70,000 | 15,000 | 5,000 | $28 \times 28$ | 0,2,4,6,8 | 1,3,5,7,9 | 3-layer MLP |
| CIFAR-10 | 60,000 | 10,000 | 5,000 | $3 \times 32 \times 32$ | 2,3,4,5,6,7 | 0,1,8,9 | ResNet-34 |
| Optdigits | 5,620 | 1,200 | 1,125 | 62 | 0,2,4,6,8 | 1,3,5,7,9 | Linear |
| Pendigits | 10,992 | 2,500 | 2,199 | 16 | 0,2,4,6,8 | 1,3,5,7,9 | Linear |

Additionally, we report corresponding versions in the experiments that utilized absolute risk correction function (CRCR-ABS) and ReLU risk correction function (CRCR-ReLU).

## 4 EXPERIMENTS

In this section, we empirically evaluate the proposed CRCR method.

### 4.1 EXPERIMENTAL SETTINGS

**Datasets** For comprehensive experimentation, we employ four popular benchmark datasets, including MNIST [12], Kuzushiji-MNIST (K-MNIST)[3], Fashion-MNIST (F-MNIST)[25] and CIFAR-10[11]. Additionally, experiments are conducted on two UCI datasets[2], including Optdigits and Pendigits. These datasets encompass more than just two labels, therefore, we categorize the class labels into positive and negative classes, effectively transforming them into binary classification datasets. Furthermore, for each dataset, we randomly selected $m\% \times n$ instances to add noise, where the noise ratio $m$ is varied over $[0, 50, 75, 100]$. As a result, in our experiments, we generate 24 synthetic datasets in total.

Furthermore, we choose different models as backbones based on the varying feature dimensions of each dataset. Specifically, for MNIST, K-MNIST and F-MNIST, we use a 3-layer multilayer perceptron (MLP) with three hidden layers of width 300 equipped with the ReLU [17] activation function and batch normalization [8]. For CIFAR-10, we train a ResNet-34 model [6] as the backbone. For all UCI datasets, we use a linear model for training. The detailed information for each dataset is presented in Table 1.

**Baseline methods** We employ seven state-of-the-art algorithms for comparison, including four Pcomp methods (*i.e.,* PcompTeacher, PcompABS, PcompReLU and PcompUnbiased) and three ConfDiff methods (*i.e.,* ConfDiffABS, ConfDiffReLU and ConfDiffUnbiased). Details of baselines are described as follows:

- Pointwise Binary Classification with **P**airwise Confidence **Comp**arisons (**Pcomp**) [4]: A weakly supervised learning method that trains a binary classifier using pairwise comparison data, composed of unlabeled data pairs where one is more likely to be positive, instead of using pointwise data. Pcomp comprises four versions: PcompTeacher, PcompABS, PcompReLU, and PcompUnbiased. We use the code provided by its authors [1].

- Binary Classification with **Conf**idence **Diff**erence (**ConfDiff**) [22]: A weakly supervised learning method that trains a binary classifier using pairwise comparison data, which consists of pairwise unlabeled data where the difference in the probabilities of being positive (confidence difference) is known. ConfDiff comprises three versions: ConfDiff-ABS, ConfDiff-ReLU, and ConfDiff-Unbiased. We utilize the publicly available code online [2].

**Implementation details** For each comparison method under every experimental configuration, we execute the code five times, employing the logistic loss function and Adam optimizer consistently. Specifically, during the training phase, each run is independently performed for 200 epochs with a batch size of 256. In balanced scenarios (*i.e.,* $\pi = 0.5$), the learning rate is set to $10^{-3}$ across all

---

[1]https://lfeng1995.github.io/codedata.html
[2]https://github.com/wwangwitsel/ConfDiff

datasets, with weight decay parameters set to $10^{-5}$ for MNIST, K-MNIST, F-MNIST, and Pendigits, $10^{-4}$ for Optdigits, and $10^{-3}$ for Pendigits. In imbalanced scenarios (*i.e.*, $\pi = 0.2$), the learning rate is set to $10^{-4}$ for MNIST and K-MNIST, and $10^{-3}$ for the remaining datasets, with weight decay parameters set to $10^{-4}$ for K-MNIST and Optdigits, and $10^{-5}$ for the remaining datasets. During the pretraining phase, each run is independently executed for 20 epochs with a batch size of 256. The learning rate and weight decay remain consistent with those in the training phase. All experiments are conducted on a server equipped with two Nvidia RTX 4090 GPUs.

## 4.2 CONSTRUCTION OF THE CONFIDENCE DIFFERENCES

In this subsection, we present the confidence differences construction method to address the challenge of fitting scenarios where precise posterior probabilities are difficult to obtain, along with a noise generation method to validate the robustness of our method under noisy conditions.

**The confidence differences construction method**. The ConfDiff method generates class posterior probabilities using a logistic regression-based probabilistic classifier trained on labeled data and calculates the confidence difference according to its definition. Although this generation method benefits comprehensive experimental analysis, it fails to accurately reflect the posterior probability distribution derived from manual annotations in real-world scenarios. Inspired by this, we incorporate an a posterior probability construction method based on outlier detection into the probabilistic classifier and computed confidence differences according to its definition to achieve a more uniform and realistic distribution. Specifically, we apply Gaussian kernel-based probability density estimation method to discrete posterior probabilities.

$$\hat{d}(\mathrm{x}_i) = \frac{1}{nh\sqrt{2\pi}} \sum_{j=1}^{n} \exp\left(-\frac{(\mathrm{x}_i - \mathrm{x}_j)^2}{2h^2}\right),$$ (11)

where $\hat{d}(\mathrm{x}_i)$ represents the estimated probability density function at instance $\mathrm{x}_i$ and $\exp\left(-\frac{(\mathrm{x}_i-\mathrm{x}_j)^2}{2h^2}\right)$ is the standard Gaussian kernel function. Furthermore, $h$ denotes the kernel bandwidth, which controls the degree of smoothing. This parameter is adaptively set based on the standard deviation of the probability distributions used in our work. We identify instances with densities below the threshold $o$ as outliers. (Notably, $o$ is also adaptively determined based on different probability density distributions. In our work, it is set at the $2nd$ percentile of the probability density to avoid filtering out too many instances.) The posterior probabilities of remaining non-outlier instances, are then rescaled to ensure a more uniform distribution within the range $[0, 1]$.

$$p(y_i = +1|\mathbf{x}_i) = \begin{cases} \mathrm{Scaling}\left(p(y_i = +1|\mathbf{x}_i)\right), & \text{if } \hat{d}(\mathrm{x}_i) \leq o \\ p(y_i = +1|\mathbf{x}_i), & \text{otherwise} \end{cases}$$ (12)

where $\mathrm{Scaling}\left(\cdot\right)$ denotes a scaling function as:

$$\mathrm{Scaling}\left(p(y_i = +1|\mathbf{x}_i)\right) = \begin{cases} \log(p(y_i = +1|\mathbf{x}_i) + \vartheta), & \text{if } p(y_i = +1|\mathbf{x}_i) \leq 0.5 \\ \log(1 - p(y_i = +1|\mathbf{x}_i) + \vartheta), & \text{otherwise} \end{cases}$$ (13)

where $\vartheta = e^{-10}$ is a smoothing parameter. Then, the confidence difference can be calculated according to its definition $c(\mathbf{x}_i, \mathbf{x}_i') = p(y_i' = +1|\mathbf{x}_i') - p(y_i = +1|\mathbf{x}_i)$.

**The noise generation method.** One straightforward method is to add noise directly to $c$. However, this method overlooks the intrinsic logic behind the original construction of $c$. We might be more interested in observing how the noise impacts the posterior probability distribution, thereby further influencing $c$ indirectly. Then, we focus on adding noise to the posterior probabilities generated by the probabilistic classifier, thereby indirectly adding noise to $c$. In the real world, individuals tend to exhibit smaller judgment biases towards more similar sample pairs, while generating larger biases towards samples with lower similarity. Therefore, White Gaussian Noise (WGN) is introduced into the posterior probabilities $p(y_i = +1|\mathbf{x}_i)$ and $p(y_i' = +1|\mathbf{x}_i')$ provided by the probabilistic classifier for the instance pair $(\mathbf{x}_i, \mathbf{x}_i')$. Then, the noisy posterior probabilities are used to generate the label confidence difference, *i.e.*, $\tilde{c}_i = \tilde{c}(\mathbf{x}_i, \mathbf{x}_i') = \tilde{p}(y_i' = +1|\mathbf{x}_i') - \tilde{p}(y_i = +1|\mathbf{x}_i)$, where

$$\tilde{p}(y_i' = +1|\mathbf{x}_i') = p(y_i' = +1|\mathbf{x}_i') + \zeta_i', \quad \zeta_i' \sim N(0, \sigma^2)$$

$$\tilde{p}(y_i = +1|\mathbf{x}_i) = p(y_i = +1|\mathbf{x}_i) + \zeta_i, \quad \zeta_i \sim N(0, \sigma^2),$$ (14)

where $\zeta_i'$ and $\zeta_i$ represent the noise offsets which follow a standard Gaussian distribution $N(0, \sigma^2)$. In our experiments, we set $\sigma = 1/3$.

Table 2: Classification accuracy of each comparing method on six datasets (mean±std) when $\pi = 0.5$, where the best performance is shown in boldface.

| $m$ | Method | MNIST | K-MNIST | F-MNIST | CIFAR-10 | Pendigits | Optdigits |
|---|---|---|---|---|---|---|---|
| 0 | PcompUnbiased | 0.815±0.007 | 0.588±0.087 | 0.813±0.066 | 0.752±0.005 | **0.775±0.018** | 0.795±0.020 |
| | PcompReLU | 0.719±0.108 | 0.692±0.012 | 0.614±0.132 | 0.794±0.009 | 0.746±0.014 | 0.766±0.038 |
| | PcompABS | 0.830±0.005 | 0.727±0.015 | 0.837±0.010 | 0.828±0.006 | 0.645±0.059 | 0.722±0.027 |
| | PcompTeacher | 0.882±0.024 | 0.708±0.008 | 0.887±0.010 | 0.812±0.010 | 0.496±0.016 | 0.507±0.067 |
| | ConfDiffUnbiased | 0.723±0.072 | 0.576±0.029 | 0.771±0.085 | 0.848±0.014 | 0.675±0.071 | 0.799±0.023 |
| | ConfDiffReLU | 0.929±0.003 | 0.771±0.025 | 0.912±0.020 | 0.848±0.014 | 0.675±0.071 | 0.799±0.023 |
| | ConfDiffABS | 0.944±0.003 | 0.825±0.011 | 0.952±0.004 | 0.848±0.014 | 0.675±0.071 | 0.799±0.023 |
| | CRCR_Unbiased | 0.777±0.034 | 0.769±0.004 | 0.921±0.009 | **0.869±0.009** | 0.756±0.006 | **0.823±0.023** |
| | CRCR_ReLU | 0.919±0.019 | 0.685±0.080 | 0.925±0.017 | **0.869±0.009** | 0.753±0.007 | **0.823±0.023** |
| | CRCR_ABS | **0.962±0.006** | **0.848±0.013** | **0.955±0.002** | **0.869±0.009** | 0.753±0.009 | **0.823±0.023** |
| 50 | PcompUnbiased | 0.814±0.050 | 0.606±0.086 | 0.855±0.061 | 0.733±0.010 | 0.760±0.020 | 0.793±0.022 |
| | PcompReLU | 0.849±0.008 | 0.722±0.003 | 0.833±0.063 | 0.810±0.008 | 0.756±0.036 | 0.772±0.017 |
| | PcompABS | 0.853±0.016 | 0.730±0.013 | 0.876±0.015 | 0.833±0.005 | 0.676±0.069 | 0.736±0.017 |
| | PcompTeacher | 0.898±0.019 | 0.723±0.018 | 0.907±0.021 | 0.812±0.007 | 0.495±0.017 | 0.503±0.068 |
| | ConfDiffUnbiased | 0.678±0.046 | 0.602±0.021 | 0.794±0.034 | 0.833±0.013 | 0.675±0.073 | 0.792±0.021 |
| | ConfDiffReLU | 0.933±0.002 | 0.766±0.020 | 0.933±0.012 | 0.836±0.014 | 0.675±0.073 | 0.792±0.021 |
| | ConfDiffABS | 0.937±0.004 | 0.819±0.007 | 0.953±0.005 | 0.834±0.013 | 0.675±0.073 | 0.792±0.021 |
| | CRCR_Unbiased | 0.845±0.043 | 0.779±0.008 | 0.928±0.001 | 0.859±0.003 | 0.759±0.029 | **0.821±0.022** |
| | CRCR_ReLU | 0.923±0.023 | 0.793±0.019 | 0.936±0.007 | **0.860±0.003** | 0.757±0.030 | **0.821±0.022** |
| | CRCR_ABS | **0.961±0.005** | **0.851±0.010** | **0.956±0.005** | **0.860±0.003** | 0.762±0.033 | **0.821±0.022** |
| 75 | PcompUnbiased | 0.849±0.010 | 0.596±0.086 | 0.832±0.129 | 0.716±0.006 | 0.754±0.028 | 0.794±0.021 |
| | PcompReLU | 0.858±0.006 | 0.728±0.013 | 0.880±0.012 | 0.820±0.008 | 0.743±0.038 | 0.783±0.018 |
| | PcompABS | 0.865±0.008 | 0.734±0.017 | 0.874±0.011 | 0.836±0.003 | 0.688±0.060 | 0.743±0.020 |
| | PcompTeacher | 0.908±0.010 | 0.735±0.013 | 0.920±0.018 | 0.813±0.008 | 0.495±0.018 | 0.501±0.069 |
| | ConfDiffUnbiased | 0.620±0.084 | 0.560±0.025 | 0.650±0.051 | 0.844±0.008 | 0.674±0.073 | 0.795±0.018 |
| | ConfDiffReLU | 0.922±0.019 | 0.778±0.008 | 0.931±0.015 | 0.843±0.009 | 0.674±0.073 | 0.795±0.018 |
| | ConfDiffABS | 0.933±0.006 | 0.817±0.009 | 0.954±0.004 | 0.844±0.009 | 0.674±0.073 | 0.795±0.018 |
| | CRCR_Unbiased | 0.797±0.075 | 0.791±0.010 | 0.926±0.010 | **0.858±0.003** | 0.723±0.033 | **0.819±0.022** |
| | CRCR_ReLU | 0.938±0.006 | 0.792±0.010 | 0.942±0.005 | **0.858±0.003** | 0.721±0.035 | **0.819±0.022** |
| | CRCR_ABS | **0.962±0.003** | **0.851±0.006** | **0.959±0.001** | **0.858±0.003** | 0.756±0.009 | **0.819±0.022** |
| 100 | PcompUnbiased | 0.832±0.051 | 0.631±0.079 | 0.897±0.013 | 0.708±0.014 | 0.735±0.024 | 0.796±0.015 |
| | PcompReLU | 0.862±0.015 | 0.726±0.012 | 0.883±0.017 | 0.827±0.004 | 0.725±0.035 | 0.787±0.019 |
| | PcompABS | 0.865±0.014 | 0.735±0.009 | 0.886±0.009 | 0.837±0.006 | 0.688±0.059 | 0.766±0.020 |
| | PcompTeacher | 0.914±0.011 | 0.738±0.020 | 0.921±0.011 | 0.812±0.010 | 0.495±0.018 | 0.499±0.070 |
| | ConfDiffUnbiased | 0.631±0.056 | 0.548±0.022 | 0.573±0.060 | 0.835±0.012 | 0.669±0.070 | 0.791±0.021 |
| | ConfDiffReLU | 0.920±0.014 | 0.769±0.008 | 0.923±0.032 | 0.834±0.012 | 0.669±0.070 | 0.791±0.021 |
| | ConfDiffABS | 0.934±0.006 | 0.812±0.004 | 0.953±0.005 | 0.835±0.012 | 0.669±0.070 | 0.791±0.021 |
| | CRCR_Unbiased | 0.860±0.081 | 0.804±0.009 | 0.910±0.030 | **0.851±0.007** | 0.751±0.008 | **0.815±0.019** |
| | CRCR_ReLU | 0.939±0.006 | 0.797±0.006 | 0.941±0.006 | **0.851±0.007** | 0.752±0.008 | **0.815±0.019** |
| | CRCR_ABS | **0.960±0.002** | **0.856±0.008** | **0.960±0.002** | 0.851±0.007 | **0.752±0.008** | **0.815±0.019** |

## 4.3 RESULT ANALYSIS

Table 2 and Table 3 present the results of all baselines on four benchmark datasets and two UCI datasets for class-balanced (*i.e.,* prior = 0.5) and class-imbalanced scenarios (*i.e.,* prior = 0.2), respectively. Accuracy is chosen as the evaluation metric, and experiments are conducted five times on all datasets, with average and variance results recorded. Overall, our method performs nearly optimally across all scenarios compared to the baseline methods, consistently achieving nearly the best results using the ABS risk correction function.

In scenarios with balanced classes, our method outperforms Pcomp by improving accuracy from 0.02 to 0.341 and surpasses ConfDiff from 0.01 to 0.387, as observed from a baseline perspective. CRCR_ABS outperforms nearly all baselines, with the only observed exception being the results of PcompUnbiased on the Pendigits dataset when no noise is added. This may be due to the fact that the Pcomp method leverages only the information that one instance is more likely to be positive than another, without requiring knowledge of the exact difference between them. The posterior probability distribution is simply reconstructed in the absence of noise, and this reconstruction function preserves the monotonic increasing relationship of the posterior probabilities, without altering the relative likelihood of positivity between instances. Moreover, compared to Pcomp and ConfDiff, our method demonstrates increasingly stable and consistent accuracy as the noise ratio increases, with notable improvements in both accuracy and standard deviation, especially when the noise ratio reaches 100%. This indicates its ability to produce more competitive results in the presence of noise interference.

In scenarios with imbalanced classes, PcompReLU and ConfDiffReLU tend to exhibit random outcomes when confronted with imbalanced data augmented with noise. This phenomenon may be attributed to the introduced noise, which significantly increases the likelihood of predictions where

Table 3: Classification accuracy of each comparing method on six datasets (mean±std) when $\pi = 0.2$, where the best performance is shown in boldface.

| $m$ | Method | MNIST | K-MNIST | F-MNIST | CIFAR-10 | Pendigits | Optdigits |
|---|---|---|---|---|---|---|---|
| 0 | PcompUnbiased | 0.744±0.037 | 0.555±0.076 | 0.748±0.047 | 0.634±0.021 | 0.820±0.025 | 0.813±0.024 |
| | PcompReLU | 0.800±0.000 | 0.800±0.000 | 0.800±0.000 | 0.802±0.003 | 0.819±0.020 | 0.816±0.007 |
| | PcompABS | 0.804±0.009 | 0.800±0.000 | 0.801±0.001 | 0.833±0.004 | 0.797±0.023 | 0.805±0.006 |
| | PcompTeacher | 0.788±0.074 | 0.695±0.046 | 0.883±0.026 | 0.813±0.020 | 0.482±0.212 | 0.684±0.097 |
| | ConfDiffUnbiased | 0.743±0.033 | 0.622±0.077 | 0.724±0.025 | 0.812±0.004 | 0.797±0.028 | 0.830±0.016 |
| | ConfDiffReLU | 0.800±0.000 | 0.800±0.000 | 0.846±0.064 | 0.800±0.000 | 0.797±0.028 | 0.830±0.016 |
| | ConfDiffABS | 0.910±0.015 | 0.841±0.014 | **0.940±0.010** | 0.800±0.001 | 0.797±0.028 | 0.830±0.016 |
| | CRCR_Unbiased | 0.816±0.043 | 0.597±0.055 | 0.886±0.009 | **0.841±0.012** | **0.823±0.005** | **0.838±0.017** |
| | CRCR_ReLU | 0.929±0.055 | 0.814±0.031 | 0.930±0.049 | 0.801±0.001 | 0.817±0.012 | 0.830±0.008 |
| | CRCR_ABS | **0.916±0.022** | **0.856±0.006** | 0.922±0.007 | 0.812±0.017 | 0.784±0.024 | 0.825±0.005 |
| 50 | PcompUnbiased | 0.742±0.015 | 0.547±0.038 | 0.768±0.070 | 0.623±0.017 | 0.818±0.025 | 0.810±0.027 |
| | PcompReLU | 0.800±0.000 | 0.801±0.002 | 0.800±0.000 | 0.801±0.003 | 0.806±0.023 | 0.821±0.007 |
| | PcompABS | 0.824±0.029 | 0.800±0.000 | 0.809±0.006 | 0.833±0.006 | 0.801±0.030 | 0.811±0.010 |
| | PcompTeacher | 0.822±0.061 | 0.707±0.062 | 0.902±0.014 | 0.797+0.033 | 0.483±0.211 | 0.682±0.096 |
| | ConfDiffUnbiased | 0.694±0.030 | 0.640±0.043 | 0.711±0.018 | 0.805±0.006 | 0.797±0.029 | 0.834±0.015 |
| | ConfDiffReLU | 0.800±0.000 | 0.800±0.000 | 0.821±0.046 | 0.800±0.001 | 0.797±0.029 | 0.834±0.015 |
| | ConfDiffABS | 0.891±0.025 | 0.818±0.012 | 0.938±0.014 | 0.801±0.002 | 0.797±0.029 | 0.834±0.015 |
| | CRCR_Unbiased | 0.794±0.043 | 0.623±0.079 | 0.880±0.016 | 0.789±0.035 | 0.795±0.025 | **0.843±0.023** |
| | CRCR_ReLU | 0.908±0.063 | 0.815±0.015 | 0.936±0.043 | 0.811±0.025 | 0.808±0.021 | 0.838±0.014 |
| | CRCR_ABS | **0.916±0.013** | **0.830±0.029** | **0.950±0.011** | **0.850±0.017** | **0.822±0.019** | 0.835±0.011 |
| 75 | PcompUnbiased | 0.753±0.031 | 0.535±0.048 | 0.775±0.059 | 0.616±0.038 | 0.817±0.017 | 0.813±0.030 |
| | PcompReLU | 0.804±0.009 | 0.804±0.007 | 0.800±0.000 | 0.805±0.012 | 0.822±0.020 | 0.827±0.008 |
| | PcompABS | 0.863±0.014 | 0.800±0.000 | 0.828±0.016 | 0.832±0.005 | 0.803±0.038 | 0.813±0.009 |
| | PcompTeacher | 0.840±0.061 | 0.714±0.055 | 0.908±0.019 | 0.793±0.044 | 0.482±0.211 | 0.680±0.096 |
| | ConfDiffUnbiased | 0.704±0.058 | 0.630±0.026 | 0.745±0.088 | 0.804±0.004 | 0.796±0.031 | 0.830±0.016 |
| | ConfDiffReLU | 0.800±0.000 | 0.800±0.000 | 0.800±0.000 | 0.800±0.000 | 0.796±0.031 | 0.830±0.016 |
| | ConfDiffABS | 0.862±0.030 | 0.806±0.005 | 0.921±0.023 | 0.800±0.001 | 0.796±0.031 | 0.830±0.016 |
| | CRCR_Unbiased | 0.792±0.047 | 0.640±0.028 | 0.861±0.033 | 0.772±0.020 | 0.811±0.030 | 0.836±0.022 |
| | CRCR_ReLU | 0.901±0.055 | 0.817±0.024 | 0.801±0.001 | 0.828±0.024 | **0.827±0.008** | 0.836±0.017 |
| | CRCR_ABS | **0.914±0.008** | **0.819±0.022** | **0.947±0.006** | **0.853±0.004** | 0.819±0.011 | **0.839±0.012** |
| 100 | PcompUnbiased | 0.752±0.021 | 0.540±0.069 | 0.834±0.034 | 0.643±0.053 | 0.805±0.024 | 0.817±0.027 |
| | PcompReLU | 0.845±0.040 | 0.808±0.010 | 0.814±0.019 | 0.806±0.005 | 0.808±0.020 | 0.834±0.008 |
| | PcompABS | 0.871±0.006 | 0.801±0.001 | 0.844±0.015 | 0.835±0.003 | 0.803±0.029 | 0.823±0.012 |
| | PcompTeacher | 0.869±0.068 | 0.711±0.062 | 0.922±0.011 | 0.787±0.033 | 0.482±0.211 | 0.681±0.096 |
| | ConfDiffUnbiased | 0.772±0.056 | 0.693±0.028 | 0.748±0.101 | 0.810±0.007 | 0.796±0.028 | 0.831±0.017 |
| | ConfDiffReLU | 0.800±0.000 | 0.800±0.000 | 0.800±0.000 | 0.800±0.001 | 0.796±0.028 | 0.831±0.016 |
| | ConfDiffABS | 0.814±0.006 | 0.801±0.002 | 0.870±0.043 | 0.801±0.001 | 0.796±0.028 | 0.831±0.016 |
| | CRCR_Unbiased | 0.790±0.036 | 0.639±0.059 | 0.838±0.056 | 0.780±0.014 | 0.797±0.018 | 0.837±0.026 |
| | CRCR_ReLU | 0.905±0.059 | 0.808±0.007 | 0.903±0.039 | 0.800±0.001 | 0.800±0.014 | 0.838±0.020 |
| | CRCR_ABS | **0.926±0.010** | **0.828±0.025** | **0.958±0.009** | **0.841±0.005** | **0.810±0.006** | **0.839±0.019** |

one instance in a pair is incorrectly predicted to be more likely positive than the other, contrary to the actual scenario. This contradiction becomes significantly more pronounced as class imbalance and noise ratio increase. For other baselines, we observe advantages in both accuracy mean and variance. From the dataset perspective, CRCR_ABS significantly outperforms other methods on the MNIST, K-MNIST, F-MNIST, and CIFAR-10 datasets in the presence of noise, while maintaining strong competitiveness on the Pendigits and Optdigits datasets. CRCR_Unbiased shows promising results without noise; however, the experiments clearly demonstrate that its training challenges on complex and noisy datasets often lead to a notable decline in performance. This further underscores the effectiveness of CRCR_ABS in maintaining robust performance when dealing with complex datasets.

### 4.4 PARAMETER SENSITIVITY

In this subsection, we conduct experiments with different thresholds $\theta$ for partitioning subsets and the parameter $\alpha$ for the consistency term, and the results are shown in Figure 2.

**About different threshold $\theta$** To evaluate the sensitivity of the threshold $\theta$, we vary its value within the range $\{0.1, 0.2, ..., 1\}$ and examine its influence on four distinct benchmark datasets (*i.e.,* MNIST, K-MNIST, F-MNIST and CIFAR-10). The results reveal that the accuracy score peaks for the four benchmark datasets when $\theta = 0.4$ with $\pi = 0.5$, and when $\theta = 0.2$ or $0.3$ with $\pi = 0.2$. This observation may be attributed to the distribution of confidence differences resembling a waveform akin to a normal distribution. A low threshold results in numerous inaccurate predictions within the subset $D^S$ utilized for risk consistency, while a high threshold leads to a scarcity of samples within

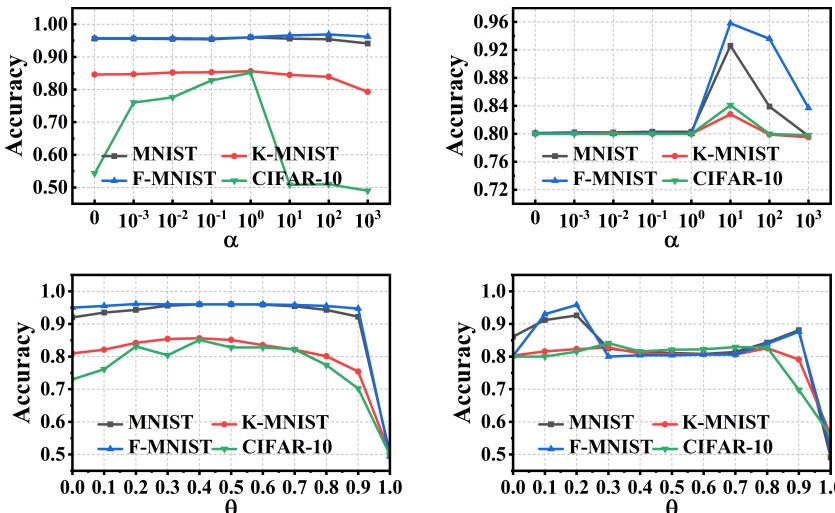

Figure 2: Sensitivity analysis of parameters $\alpha$ (top) and $\theta$ (bottom) on four benchmark datasets when $\pi = 0.5$ (left) and $\pi = 0.2$ (right).

$D^S$, thus diminishing the available supervisory information. Therefore, we empirically recommend setting the threshold at $\theta = 0.4$ when $\pi = 0.5$, and $\theta = 0.2$ or $0.3$ when $\pi = 0.2$.

**About different parameter** $\alpha$  To assess the sensitivity of the parameter $\alpha$, we vary its values across the range $\{10^i | i = -3, \ldots, +3\}$ and observe its effects on four benchmark datasets. Our analysis reveals that $\alpha$ shows increased sensitivity on the larger-scale CIFAR-10 dataset when $\pi = 0.5$, while maintaining relatively stable performance on the smaller-scale datasets. Moreover, $\alpha$ leads to a consistent trend in accuracy variation across the four datasets when $\pi = 0.2$. Notably, it achieves relatively optimal results when $\alpha = 1$ with $\pi = 0.5$, and $\alpha = 10^1$ with $\pi = 0.2$. Thus, we recommend setting $\alpha = 1$ or $10^1$ in experimental setups.

### 4.5 ABLATION STUDY

In this subsection, we conduct ablation studies on various strategies by setting corresponding parameters to zero. Specifically, setting $\{\alpha = 0, \theta = 0\}$ represent versions without consistency strategy and without subset segmentation strategy, respectively. The experimental results, presented in Figure 1, demonstrate that our proposed subset segmentation strategy and consistency term contribute to performance improvement to some extent in the context of noisy confidence difference classification.

## 5 CONCLUSION

In this paper, we propose a novel ConfDiff classification method based on consistency risk and consistency regularization to address the challenge of noisy supervised signals in ConfDiff classification. We conduct a theoretical analysis of various supervised signals associated with different confidence differences. Based on this analysis, the ConfDiff dataset is partitioned into two subsets according to the reliability of the supervised information. For the subset with more reliable supervision, we employ a consistency risk to preserve precise supervised information. Conversely, for the subset with less reliable supervision, we leverage consistency regularization to mitigate the impact of erroneous predictions. Extensive experimental results demonstrate that the proposed CRCR method outperforms state-of-the-art baselines and exhibits strong robustness, even under artificially induced noise.

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

## A  PROOF OF THEOREM 1

In this appendix, we provide the proof of the Theorem 1 and the corresponding technical lemmas.

**Lemma 1.** *The Rademacher complexity $\bar{\Re}_n(\mathcal{L}_{\mathrm{CRCR}} \circ \mathcal{G})$ on $\mathcal{D}$ for ConfDiff data with noise of size $n$ can be defined as follows:*

$$\bar{\Re}_n(\mathcal{L}_{\mathrm{CRCR}} \circ \mathcal{G}) \leq 2L_\ell \Re_{n_1}(\mathcal{G}) + \frac{\alpha}{\log(\varepsilon)} \Re_{n_2}(\mathcal{G}) \tag{15}$$

The proof of Lemma 1:

$$\begin{aligned}
\bar{\Re}_n(\mathcal{L}_{\mathrm{CRCR}} \circ \mathcal{G}) =& \mathbb{E}_{\mathcal{D}_{n_1}} \mathbb{E}_\sigma [\sup_{g \in \mathcal{G}} \frac{1}{n_1} \sum_{i=1}^{n_1} \sigma_i \mathcal{L}_{\mathrm{CRCR}}^S(g; \mathbf{x}_i, \mathbf{x}_i')] \\
&+ \mathbb{E}_{\mathcal{D}_{n_2}} \mathbb{E}_\sigma [\sup_{g \in \mathcal{G}} \frac{1}{n_2} \sum_{i=1}^{n_2} \sigma_i \mathcal{L}_{\mathrm{CRCR}}^C(g; \mathbf{x}_i, \mathbf{x}_i')] \\
=& \mathbb{E}_{\mathcal{D}_{n_1}} \mathbb{E}_\sigma [\sup_{g \in \mathcal{G}} \frac{1}{n_1} \sum_{i=1}^{n_1} \frac{1}{2} \sigma_i((\pi - c_i)\ell(g(\mathbf{x}_i), +1) + (1 - \pi - c_i)\ell(g(\mathbf{x}_i'), -1) \\
&\qquad\qquad\qquad + (\pi + c_i)\ell(g(\mathbf{x}_i'), +1) + (1 - \pi + c_i)\ell(g(\mathbf{x}_i), -1))] \\
&+ \mathbb{E}_{\mathcal{D}_{n_2}} \mathbb{E}_\sigma [\sup_{g \in \mathcal{G}} \frac{1}{n_2} \sum_{i=1}^{n_2} \alpha \sigma_i \frac{1}{\log(|\tilde{c}_i| + \varepsilon)} \cdot \|(g(\mathbf{x}_i) - g(\mathbf{x}_i')\|_2] \\
=& \mathbb{E}_{\mathcal{D}_{n_1}} \mathbb{E}_\sigma [\sup_{g \in \mathcal{G}} \frac{1}{n_1} \sum_{i=1}^{n_1} \sigma_i \left\| \bigtriangledown \mathcal{L}_{\mathrm{CD}}^S(g; \mathbf{x}_i, \mathbf{x}_i') \right\|_2 g(\mathbf{x}_i)] \\
&+ \mathbb{E}_{\mathcal{D}_{n_2}} \mathbb{E}_\sigma [\sup_{g \in \mathcal{G}} \frac{1}{n_2} \sum_{i=1}^{n_2} \sigma_i \left\| \bigtriangledown \mathcal{L}_{\mathrm{CD}}^C(g; \mathbf{x}_i, \mathbf{x}_i') \right\|_2 g(\mathbf{x}_i)]
\end{aligned} \tag{16}$$

where

$$\begin{aligned}
& \left\| \bigtriangledown \mathcal{L}_{\mathrm{CRCR}}^S(g; \mathbf{x}_i, \mathbf{x}_i') \right\|_2 \\
=& \frac{1}{2} \left\| \bigtriangledown ((\pi - c_i)\ell(g(\mathbf{x}_i), +1) + (1 - \pi - c_i)\ell(g(\mathbf{x}_i'), -1) \right. \\
& \left. + (\pi + c_i)\ell(g(\mathbf{x}_i'), +1) + (1 - \pi + c_i)\ell(g(\mathbf{x}_i), -1)) \right\|_2 \\
\leq& \frac{1}{2} \Big( \left\| \bigtriangledown ((\pi - c_i)\ell(g(\mathbf{x}_i), +1)) \right\|_2 + \left\| \bigtriangledown ((1 - \pi - c_i)\ell(g(\mathbf{x}_i'), -1)) \right\|_2 \\
& + \left\| \bigtriangledown ((\pi + c_i)\ell(g(\mathbf{x}_i'), +1)) \right\|_2 + \left\| \bigtriangledown ((1 - \pi + c_i)\ell(g(\mathbf{x}_i), -1)) \right\|_2 \Big) \\
\leq& \frac{1}{2} |\pi - c_i| L_\ell + \frac{1}{2} |1 - \pi - c_i| L_\ell + \frac{1}{2} |\pi + c_i| L_\ell + \frac{1}{2} |1 - \pi + c_i| L_\ell \\
\leq& 2L_\ell
\end{aligned} \tag{17, 18}$$

and,

$$\begin{aligned}
\left\| \bigtriangledown \mathcal{L}_{\mathrm{CRCR}}^S(g; \mathbf{x}_i, \mathbf{x}_i') \right\|_2 =& \alpha \left\| \bigtriangledown \frac{1}{\log(|\tilde{c}_i| + \varepsilon)} \cdot \|(g(\mathbf{x}_i) - g(\mathbf{x}_i')\|_2 \right\|_2 \\
\leq& \alpha \frac{1}{\log(|\tilde{c}_i| + \varepsilon)} \cdot \frac{g(\mathbf{x}_i) - g(\mathbf{x}_i')}{\|g(\mathbf{x}_i) - g(\mathbf{x}_i')\|_2} \\
\leq& \frac{\alpha}{\log(\varepsilon)}
\end{aligned} \tag{19}$$

Replacing the corresponding term in Eq.16 with Eq.18 and Eq.19, we can prove the Lemma 1:

$$\begin{aligned}
\bar{\Re}_n(\mathcal{L}_{\mathrm{CRCR}} \circ \mathcal{G}) \leq& 2L_\ell \mathbb{E}_{\mathcal{D}_{n_1}} \mathbb{E}_\sigma [\sup_{g \in \mathcal{G}} \frac{1}{n_1} \sum_{i=1}^{n_1} \sigma_i g(\mathbf{x}_i)] + \frac{\alpha}{\log(\varepsilon)} \mathbb{E}_{\mathcal{D}_{n_2}} \mathbb{E}_\sigma [\sup_{g \in \mathcal{G}} \frac{1}{n_2} \sum_{i=1}^{n_2} \sigma_i g(\mathbf{x}_i)] \\
\leq& 2L_\ell \Re_{n_1}(\mathcal{G}) + \frac{\alpha}{\log(\varepsilon)} \Re_{n_2}(\mathcal{G})
\end{aligned} \tag{20}$$

**Lemma 2.**

$$\sup_{g \in \mathcal{G}} \left| R(g) - \hat{R}_{\mathrm{CRCR}}(g) \right| \leq 4L_\ell \mathfrak{R}_{n_1}(\mathcal{G}) + \frac{2\alpha}{\log(\varepsilon)} \mathfrak{R}_{n_2}(\mathcal{G})$$

$$+ \left( \frac{C_\ell}{n_1} + \left| \frac{1}{\log(\varepsilon)} - \frac{1}{\log(\theta + \varepsilon)} \right| \frac{4\alpha C_g{}^2}{n_2} \right) \sqrt{2n\mathrm{In}(2/\delta)} \quad (21)$$

The proof of Lemma 2: Let $\hat{R}_{\mathrm{CRCR}}(g)$ and $\hat{\tilde{R}}_{\mathrm{CRCR}}(g)$ represent the empirical risks of two sets of training samples, each differing by exactly one point, denoted as $\{(\mathbf{x}_i, \mathbf{x}'_i), c(\mathbf{x}_i, \mathbf{x}'_i)\}$ and $\{(\bar{\mathbf{x}}_i, \bar{\mathbf{x}}'_i), c(\bar{\mathbf{x}}_i, \bar{\mathbf{x}}'_i)\}$ respectively.

$$\sup_{g \in \mathcal{G}} \left| \left( R(g) - \hat{\tilde{R}}_{\mathrm{CRCR}}(g) \right) - \left( R(g) - \hat{R}_{\mathrm{CRCR}}(g) \right) \right|$$

$$\leq \sup_{g \in \mathcal{G}} \left| \hat{R}_{\mathrm{CRCR}}(g) - \hat{\tilde{R}}_{\mathrm{CRCR}}(g) \right|$$

$$\leq \sup_{g \in \mathcal{G}} \left| \frac{1}{2n_1}(\pi - \tilde{c}_i)\Big(\ell\big(g(\mathbf{x}_i), +1\big) - \ell\big(g(\bar{\mathbf{x}}_i), +1\big)\Big) \right. \quad (22)$$

$$+ (1 - \pi - \tilde{c}_i)\Big(\ell\big(g(\mathbf{x}'_i), -1\big) - \ell\big(g(\bar{\mathbf{x}}'_i), -1\big)\Big) \quad (23)$$

$$+ (\pi + \tilde{c}_i)\Big(\ell\big(g(\mathbf{x}'_i), +1\big) - \ell\big(g(\bar{\mathbf{x}}'_i), +1\big)\Big) \quad (24)$$

$$+ (1 - \pi + \tilde{c}_i)\Big(\ell\big(g(\mathbf{x}_i), -1\big) - \ell\big(g(\bar{\mathbf{x}}_i), -1\big)\Big) \quad (25)$$

$$\left. + \frac{\alpha}{n_2}\left( \frac{1}{\log(|\tilde{c}_i| + \varepsilon)} \cdot \|(g(\mathbf{x}_i) - g(\mathbf{x}'_i)\|_2 - \frac{1}{\log(|\tilde{\bar{c}}_i| + \varepsilon)} \cdot \|(g(\bar{\mathbf{x}}_i) - g(\bar{\mathbf{x}}'_i)\|_2 \right) \right|$$

$$\leq \frac{2C_\ell}{n_1} + \left| \frac{1}{\log(\varepsilon)} - \frac{1}{\log(\theta + \varepsilon)} \right| \frac{2\alpha C_g}{n_2} \quad (26)$$

Then according McDiarmid's inequality:

$$\sup_{g \in \mathcal{G}} \left| R(g) - \hat{R}_{\mathrm{CRCR}}(g) \right| \leq \mathbb{E}_{\mathcal{D}_n}[\sup_{g \in \mathcal{G}}\big(R(g) - \hat{R}_{\mathrm{CRCR}}(g)\big)]$$

$$+ \left( \frac{2C_\ell}{n_1} + \left| \frac{1}{\log(\varepsilon)} - \frac{1}{\log(\theta + \varepsilon)} \right| \frac{2\alpha C_g}{n_2} \right) \sqrt{2n\mathrm{In}(2/\delta)}$$

$$\leq 2\bar{\mathfrak{R}}_n(\mathcal{L}_{\mathrm{CRCR}} \circ \mathcal{G})$$

$$+ \left( \frac{2C_\ell}{n_1} + \left| \frac{1}{\log(\varepsilon)} - \frac{1}{\log(\theta + \varepsilon)} \right| \frac{2\alpha C_g}{n_2} \right) \sqrt{2n\mathrm{In}(2/\delta)}$$

$$\leq 4L_\ell \mathfrak{R}_{n_1}(\mathcal{G}) + \frac{2\alpha}{\log(\varepsilon)} \mathfrak{R}_{n_2}(\mathcal{G})$$

$$+ \left( \frac{2C_\ell}{n_1} + \left| \frac{1}{\log(\varepsilon)} - \frac{1}{\log(\theta + \varepsilon)} \right| \frac{2\alpha C_g}{n_2} \right) \sqrt{2n\mathrm{In}(2/\delta)} \quad (27)$$

The proof of Theorem 1:

$$R(\hat{g}_{\mathrm{CRCR}}) - R(g^*) = \big(R(\hat{g}_{\mathrm{CRCR}}) - \hat{R}_{\mathrm{CRCR}}(\hat{g}_{\mathrm{CRCR}})\big) + \big(\hat{R}_{\mathrm{CRCR}}(\hat{g}_{\mathrm{CRCR}}) - \hat{R}_{\mathrm{CRCR}}(g^*)\big)$$

$$+ \big(\hat{R}_{\mathrm{CRCR}}(g^*) - R(g^*)\big)$$

$$\leq \big(R(\hat{g}_{\mathrm{CRCR}}) - \hat{R}_{\mathrm{CRCR}}(\hat{g}_{\mathrm{CRCR}})\big) + \big(\hat{R}_{\mathrm{CRCR}}(g^*) - R(g^*)\big)$$

$$\leq 2 \sup_{g \in \mathcal{G}} \left| R(g) - \hat{R}_{\mathrm{CRCR}}(g) \right|$$

$$\leq 8L_\ell \mathfrak{R}_{n_1}(\mathcal{G}) + \frac{4\alpha}{\log(\varepsilon)} \mathfrak{R}_{n_2}(\mathcal{G})$$

$$+ \left( \frac{4C_\ell}{n_1} + \left| \frac{1}{\log(\varepsilon)} - \frac{1}{\log(\theta + \varepsilon)} \right| \frac{4\alpha C_g}{n_2} \right) \sqrt{2n\mathrm{In}(2/\delta)} \quad (28)$$

## B    PROOF OF EQ. 6

In this appendix, we provide the proof of the Eq. 6.

Substituting the form of the loss function from Eq.5 into Eq.3, then we can obtain:

$$
\begin{aligned}
R_{\mathrm{CD}}(g) &= \frac{1}{2}\mathbb{E}_{p(\mathbf{x},\mathbf{x}')}\Big[\big(\pi - c(\mathbf{x},\mathbf{x}')\big)\ell\big(g(\mathbf{x}),+1\big) + \big(1 - \pi - c(\mathbf{x},\mathbf{x}')\big)\ell\big(g(\mathbf{x}'),-1\big) \\
&\qquad\qquad + \big(\pi + c(\mathbf{x},\mathbf{x}')\big)\ell\big(g(\mathbf{x}'),+1\big) + \big(1 - \pi + c(\mathbf{x},\mathbf{x}')\big)\ell\big(g(\mathbf{x}),-1\big)\Big] \\
&= \frac{1}{2}\mathbb{E}_{p(\mathbf{x},\mathbf{x}')}\Big[\big(\pi - c(\mathbf{x},\mathbf{x}')\big)\big(h(g(\mathbf{x})) - g(\mathbf{x})\big) + \big(1 - \pi - c(\mathbf{x},\mathbf{x}')\big)\big(h(g(\mathbf{x}')) + g(\mathbf{x}')\big) \\
&\qquad\qquad + \big(\pi + c(\mathbf{x},\mathbf{x}')\big)\big(h(g(\mathbf{x}')) - g(\mathbf{x}')\big) + \big(1 - \pi + c(\mathbf{x},\mathbf{x}')\big)\big(h(g(\mathbf{x})) + g(\mathbf{x})\big)\Big] \\
&= \frac{1}{2}\mathbb{E}_{p(\mathbf{x},\mathbf{x}')}\Big[h\big(g(\mathbf{x})\big) + \big(1 - 2\pi + 2c(\mathbf{x},\mathbf{x}')\big)g(\mathbf{x}) \\
&\qquad\qquad + h\big(g(\mathbf{x}')\big) + \big(1 - 2\pi - 2c(\mathbf{x},\mathbf{x}')\big)g(\mathbf{x}')\Big] \\
&= \frac{1}{2}\mathbb{E}_{p(\mathbf{x},\mathbf{x}')}\Big[h\big(g(\mathbf{x})\big) + 2c(\mathbf{x},\mathbf{x}')g(\mathbf{x}) + h\big(g(\mathbf{x}')\big) - 2c(\mathbf{x},\mathbf{x}')g(\mathbf{x}')\Big] \\
&\quad + \frac{1}{2}\mathbb{E}_{p(\mathbf{x},\mathbf{x}')}\Big[(1 - 2\pi)\big(g(\mathbf{x}) + g(\mathbf{x}')\big)\Big] \\
&= \frac{1}{2}\mathbb{E}_{p(\mathbf{x},\mathbf{x}')}\Big[h\big(g(\mathbf{x})\big) + 2c(\mathbf{x},\mathbf{x}')g(\mathbf{x}) + h\big(g(\mathbf{x}')\big) - 2c(\mathbf{x},\mathbf{x}')g(\mathbf{x}') \\
&\qquad\qquad + c(\mathbf{x},\mathbf{x}')h\big(g(\mathbf{x})\big) - c(\mathbf{x},\mathbf{x}')h\big(g(\mathbf{x})\big) + \frac{1}{2}g(\mathbf{x}) - \frac{1}{2}g(\mathbf{x}) \\
&\qquad\qquad + c(\mathbf{x},\mathbf{x}')h\big(g(\mathbf{x}')\big) - c(\mathbf{x},\mathbf{x}')h\big(g(\mathbf{x}')\big) + \frac{1}{2}g(\mathbf{x}') - \frac{1}{2}g(\mathbf{x}')\Big] \\
&\quad + \frac{1}{2}\mathbb{E}_{p(\mathbf{x},\mathbf{x}')}\Big[(1 - 2\pi)\big(g(\mathbf{x}) + g(\mathbf{x}')\big)\Big] \\
&= \frac{1}{2}\mathbb{E}_{p(\mathbf{x},\mathbf{x}')}\Big[\frac{1}{2}h\big(g(\mathbf{x})\big) - c(\mathbf{x},\mathbf{x}')h\big(g(\mathbf{x})\big) - \frac{1}{2}g(\mathbf{x}) + c(\mathbf{x},\mathbf{x}')g(\mathbf{x}) \\
&\qquad\qquad + \frac{1}{2}h\big(g(\mathbf{x}')\big) + c(\mathbf{x},\mathbf{x}')h\big(g(\mathbf{x}')\big) - \frac{1}{2}g(\mathbf{x}') - c(\mathbf{x},\mathbf{x}')g(\mathbf{x}')\Big] \\
&\quad + \frac{1}{2}\mathbb{E}_{p(\mathbf{x},\mathbf{x}')}\Big[(1 - 2\pi)\big(g(\mathbf{x}) + g(\mathbf{x}')\big)\Big] \\
&= \frac{1}{2}\mathbb{E}_{p(\mathbf{x},\mathbf{x}')}\Big[\frac{1}{2}h\big(g(\mathbf{x})\big) - c(\mathbf{x},\mathbf{x}')h\big(g(\mathbf{x})\big) - \frac{1}{2}g(\mathbf{x}) + c(\mathbf{x},\mathbf{x}')g(\mathbf{x}) \\
&\qquad\qquad + \frac{1}{2}h\big(g(\mathbf{x}')\big) + c(\mathbf{x},\mathbf{x}')h\big(g(\mathbf{x}')\big) - \frac{1}{2}g(\mathbf{x}') - c(\mathbf{x},\mathbf{x}')g(\mathbf{x}')\Big] \\
&\quad + \frac{1}{2}\mathbb{E}_{p(\mathbf{x},\mathbf{x}')}\Big[\frac{1}{2}h\big(g(\mathbf{x})\big) + c(\mathbf{x},\mathbf{x}')h\big(g(\mathbf{x})\big) + \frac{1}{2}g(\mathbf{x}) + c(\mathbf{x},\mathbf{x}')g(\mathbf{x}) \\
&\qquad\qquad + \frac{1}{2}h\big(g(\mathbf{x}')\big) - c(\mathbf{x},\mathbf{x}')h\big(g(\mathbf{x}')\big) + \frac{1}{2}g(\mathbf{x}') - c(\mathbf{x},\mathbf{x}')g(\mathbf{x}')\Big] \\
&\quad + \frac{1}{2}\mathbb{E}_{p(\mathbf{x},\mathbf{x}')}\Big[(1 - 2\pi)\big(g(\mathbf{x}) + g(\mathbf{x}')\big)\Big] \\
&= \frac{1}{2}\mathbb{E}_{p(\mathbf{x},\mathbf{x}')}\Big[(\frac{1}{2} - c(\mathbf{x},\mathbf{x}'))\ell\big(g(\mathbf{x}),+1\big) + (\frac{1}{2} + c(\mathbf{x},\mathbf{x}'))\ell\big(g(\mathbf{x}'),+1\big)\Big] \\
&\quad + \frac{1}{2}\mathbb{E}_{p(\mathbf{x},\mathbf{x}')}\Big[(\frac{1}{2} + c(\mathbf{x},\mathbf{x}'))\ell\big(g(\mathbf{x}),-1\big) + (\frac{1}{2} - c(\mathbf{x},\mathbf{x}'))\ell\big(g(\mathbf{x}'),-1\big)\Big] \\
&\quad + \frac{1}{2}\mathbb{E}_{p(\mathbf{x},\mathbf{x}')}\Big[(1 - 2\pi)\big(g(\mathbf{x}) + g(\mathbf{x}')\big)\Big].
\end{aligned}
\tag{29}
$$

Then Eq. 6 is proven.

## C    LIMITATIONS

The noise generation method we proposed primarily utilizes a Gaussian distribution to perturb confidence difference distributions originally concentrated around specific values, aiming to approximate the confidence difference distributions that may manifest in the real world. Consequently, artificial datasets are utilized. In the future, we may consider annotating pairwise confidence difference datasets derived from real-world scenarios. It would allow for experiments using authentic datasets rather than artificially constructed ones, offering substantial practical significance.

Additionally, the datasets used are actually multi-label datasets although we focus on binary classification problems in weakly supervised learning. Then the labels of these multi-label datasets are partitioned into two disjoint subsets, each serving as positive and negative classes, respectively, thereby converting them into binary classification datasets. In the future, we will consider expanding the problem scenario to multi-label classification.

## D    BROADER IMPACTS

The noise confidence difference classification proposed in this paper stands to notably improve decision accuracy in real-world settings. It addresses potential noise impacts present in real-world data and holds substantial practical significance as a plausible scenario in weakly supervised domains. Its applicability can be extended to various fields including medical diagnosis, rehabilitation assessment, and financial risk management.

However, it's important to acknowledge that the confidence difference utilized in our method within weakly supervised settings might be influenced by potential data biases inherent in the real world. Furthermore, we demonstrate the effectiveness of our approach in weakly supervised scenarios, there's a risk of excessive dependence on algorithms for decision-making, potentially overlooking the cultivation of individual decision-making capabilities and autonomy.

