# OpenReview forum: "Confidence Difference Reflects Various Supervised Signals in Confidence-Difference Classification"
_ICLR.cc/2025/Conference — Submitted to ICLR 2025_

### Official Review · Reviewer_R1j9 · 2024-10-18

**Soundness:** 4
**Presentation:** 3
**Contribution:** 4
**Rating:** 8
**Confidence:** 4

**Summary:**

This paper deals with confidence-difference classification, a weakly supervised binary classification problem. In order to mitigate the noise contained in the confidence differences, a novel risk estimator using consistency regularization is employed to improve performance. Extensive experiments on benchmark datasets validate the effectiveness of the proposed method.

**Strengths:**

- The problem studied, i.e., confidence-difference classification with noise, is an interesting and promising problem.
- The motivation that different confidence difference values convey different information is valid and interesting, which has not been explored in the literature.
- The proposed consistency regularization term is valid, which can improve the performance when the confidence difference is small for some training data.
- The experiments are comprehensive and validation studies are also conducted to validate the effectiveness of the proposed method.
- The paper is clearly written and easy to understand.

**Weaknesses:**

- One of my concerns is whether the noise is more pronounced when the confidence difference is small. When the confidence difference is large, noise can also be introduced. Although it does not affect the algorithm, which I think can work well even if no noise is introduced in the experiments. I think this point needs clarification and the paper can be polished as well. I think a more reasonable story is to use some terms like "clear" and "ambiguous" to replace data with and without noise. Some data pairs are more clear (larger confidence difference absolute values) and their labels can be (+1,-1) or (-1,+1) with a high probability. Some data are ambiguous (small confidence differences) and we cannot tell if they are both positive or negative. This story may be more reasonable from my point of view. To handle different types of data, the consistency regularization term is introduced to improve learning from ambiguous data pairs.

- The descriptions in Section 4.2 are not so clear. I cannot understand how the data is generated, such as the meaning of $c_{non-outlier}$.

- There are some unclear notations and expressions that should be examined. For example, $\ell$ is missing in line 235. In line 241 it should be $\hat{g}$ instead of $R(\hat{g})$. In line 42-43, it seems to apply to general supervised learning instead of weak supervised learning? In line 89 it should read "difference". Also, more discussion should be added after Theorem 1. The space for the post text can be reduced. Also, the "\_" notation for the methods can be replaced with "-".

**Questions:**

- Why do larger confidence differences contain less noise?

---

> ### Author Response · Authors · 2024-11-22
> **Response for Reviewer R1j9**
>
> **Q1. Whether noise is more pronounced with small confidence differences, as large confidence differences can also introduce noise. And it can work well even if no noise is introduced in the experiments.**
>
> Thank you for your suggestions. According to the analysis in "**General Response: Q1. About the motivation of this paper**", $R_{CD}$ encourages $\mathbf{x}$ and $\mathbf{x'}$ to adjust their predictions in opposite directions. This prediction trend holds correctly for pairwise instances with $\left|c(\mathbf{x}, \mathbf{x}')\right| > 0.5$. However, when $\left|c(\mathbf{x}, \mathbf{x}')\right| \le 0.5$, $\mathbf{x}$ and $\mathbf{x'}$ may belong to the same class or different classes. In such cases, the prediction trend may lead to samples from the same class being predicted as belonging to different classes, introducing erroneous supervisory signals. So noise is more pronounced with small confidence differences.
>
> Moreover, our experimental results show that our method outperforms other baselines even in the absence of noise (i.e., noise ratio $m=0$). Furthermore, when the noise ratio is high, our method still demonstrates strong robustness.
>
> &nbsp;
>
> **Q2. The descriptions in Section 4.2 are not so clear, particularly regarding data generation.**
>
> Many thanks for your valuable comments. Firstly, in the pretraining phase, we train a probabilistic classifier using standard labeled data through logistic regression. Then, we input randomly pairwise unlabeled data into the classifier to generate class posterior probabilities. Subsequently, we discard misclassified data to ensure that no noise arises from calibration errors of the probabilistic classifier during posterior generation.
>
> Secondly, we apply a Gaussian-kernel-based probability density estimation method to detect outliers in the posterior probability distribution. This step primarily aims to prevent localized density concentration in the posterior distribution during subsequent scaling due to outlier instances.
>
> Thirdly, we retain the original posterior probability values for the outliers identified in the previous step and apply a designed scaling function to the remaining non-outlier instances. This approach ensures that the reconstructed posterior probabilities form a more uniform distribution.
>
> Finally, Gaussian white noise is added to the scaled posterior probabilities, indirectly introducing noise into the confidence difference and yielding the final $ \tilde{c} $.
>
> &nbsp;
>
> **Q3. There are some unclear notations and expressions that should be examined.**
>
> Thank you for your corrections. The error in lines 89, 235, and 241 will be revised in future versions.
>
> **About pointwise information in line 42-43.** Pointwise information is also present in weakly supervised learning beyond general supervised learning. In approaches like Soft Label Learning and Confidence Learning, models receive probabilities or confidence scores rather than binary labels. Maximizing the use of such pointwise weakly supervised information is a key direction in weakly supervised learning, driving advancements in techniques like soft labeling, mixup, and others.
>
> **About discussion for Theorem 1.** Please refer to "**General Response: Q2. About the analysis for Theorem 1**".
>
> &nbsp;
>
> **Q4. Why do larger confidence differences contain less noise?**
>
> Thank you very much for your valuable suggestions. According to the analysis in "**General Response: Q1. About the motivation of this paper**", the confidence difference is defined as $ c(\mathbf{x}\_i,\mathbf{x}'\_i) = p(y'\_i=+1|\mathbf{x}'\_i) - p(y\_i=+1|\mathbf{x}\_i) \in [-1,1]$. If $\left|c(\mathbf{x}, \mathbf{x}')\right| > 0.5$, $\mathbf{x}$ and $\mathbf{x'}$ must belong to different classes, as the posterior difference is must surpass the classification threshold. Additionally, $R_{CD}$ encourages $\mathbf{x}$ and $\mathbf{x'}$ to adjust their predictions in opposite directions. This prediction trend holds correctly for pairwise instances with $\left|c(\mathbf{x}, \mathbf{x}')\right| > 0.5$. Therefore, we can deduce that larger confidence differences contain less noise.

---

> > ### Comment · Reviewer_R1j9 · 2024-11-26
> >
> > Thanks for the reply. I will keep my score.

---

### Official Review · Reviewer_wT35 · 2024-10-28

**Soundness:** 3
**Presentation:** 2
**Contribution:** 3
**Rating:** 6
**Confidence:** 3

**Summary:**

The authors study a challenging weakly supervised task called confidence difference (ConfDiff) classification, where only the posterior probability differences for pairwise data points are accessible during training. To handle the influence of noise in ConfDiff, the authors introduce a method called CRCR, which partitions the training data into two subsets with different ConfDiff levels and applies a different strategy to each subset. Additionally, theoretical analyses of the proposed methods and experimental results on multiple benchmark datasets are provided.

**Strengths:**

1. This article treats ConfDiff as a soft label problem. It partitions the dataset using a threshold and applies different strategies to handle data with high and low ConfDiff levels. The effectiveness of this method in combating noise is verified through experiments.
2. The authors strengthen the analysis by conducting ablation studies to clarify the contributions of dataset partitioning and the consistency regularization term. These experiments significantly enhance the clarity and depth of the results, making it easier to understand the individual impact of each component.
3. Theoretical analyses are provided to demonstrate the theoretical guarantees of the proposed methods.

**Weaknesses:**

1.	There is a lack of descriptions of Theorem 1.
2.	It is recommended that the authors provide a detailed explanation of 'consistency' within the text.
3.	The manuscript requires careful revision to address various issues that currently detract from its clarity and academic rigor. For example, please consider revising the following sections of the manuscript to enhance clarity and accuracy:

* line 014: Consider revising the phrase ‘’significant research significance‘’ to avoid redundancy and enhance clarity. Perhaps ‘’considerable practical significance‘’ would be more appropriate.
* line 155 Eq. (3): The notation [L(x, x), L(x, x)] may need to be corrected to [L(x, x) + L(x, x)].
* Line 174: It might be appropriate to ‘substitute the form of the loss function from Equation 5 into Equation 3.’.
* Line 210: Please clarify the meaning of $\theta(D^S)$ and $\theta(D^C)$.
* Line 241: the term $R(\hat{g}{CRCR})$ should be corrected to $\hat{g}_{CRCR}$.
* Table 3, line 387: It appears there is a labeling error, as CRCR_RelLU (0.930) is noted to significantly outperform ConfDiffABS (0.940), which seems counterintuitive. Please verify and correct.

**Questions:**

see weaknesses

---

> ### Author Response · Authors · 2024-11-22
> **Response for Reviewer wT35**
>
> **Q1. There is a lack of descriptions of Theorem 1.**
>
> Thank you for your comment. Please refer to "**General Response: Q2. About the analysis for Theorem 1**".
>
> &nbsp;
>
> **Q2. It is recommended that the authors provide a detailed explanation of 'consistency' within the text.**
>
> Thank you for your suggestions. In this work, consistency specifically refers to the expectation that, for a given instance pair $(\mathbf{x}, \mathbf{x}') \in D^{C}$, smaller values of $\left|c(\mathbf{x}, \mathbf{x}')\right|$ correspond to more similar model outputs $g(\mathbf{x})$ and $g(\mathbf{x}')$. Based on this, we design a consistency regularization term that aligns the confidence difference with the model outputs, aiming to reduce the ambiguity caused by small confidence differences. Furthermore, the motivation for this consistency regularization term is elaborated in the "**General Response: Q1. About the motivation of this paper**".
>
> &nbsp;
>
> **Q3. The manuscript needs revision to improve its clarity and academic rigor.**
>
> Thank you for your corrections. The error in lines 014, 155, 241, and 387 will be revised in future versions.
>
> About Line 210: Please clarify the meaning of $\theta(D^C) $ and $\theta(D^S) $.**
> The expression in line 210 may be confusing, as it does not define $\theta(D^C) $ and $\theta(D^S) $ as new forms; rather, $\theta$ and $D^C$ are two separate symbols. This statement aims to clarify that we partition $D$ into two subsets based on the degree of prediction error, such that $D = D^S \cup D^C $: subset $ D^S $ contains pairwise data with confidence differences satisfying $ |c(\mathbf{x}, \mathbf{x'})| > \theta $, while subset $ D^C $ includes pairwise data with confidence differences satisfying $ |c(\mathbf{x}, \mathbf{x'})| \leq \theta $.

---

> ### Author Response · Authors · 2024-11-28
> **A more detailed response for Q2.**
>
> **Q2. It is recommended that the authors provide a detailed explanation of 'consistency' within the text.**
>
> Thank you for your suggestions. Building on our previous discussion, we provide a more detailed explanation of 'consistency' from the following aspects.
>
> **What is consistency regularization?** The consistency regularization term used in this paper is formally defined as $\bigl(\frac{1}{ \log\left(\left| c(\mathbf{x},\mathbf{x}')\right| + \varepsilon  \right)  } \bigr)  \cdot  \left \| g(\mathbf{x})-g(\mathbf{x}') \right \|_2$ on $D^{C}$, aiming to encourage the classifier to further reduce output differences when the confidence difference is small. In other words, if the confidence difference between instances \(\mathbf{x}\) and \(\mathbf{x}'\) is small, \( g(\mathbf{x})\) should be similar to \( g(\mathbf{x}')\), and vice versa. This strategy helps enhance generalization ability and makes the model more robust to the noise and perturbations.
>
> **The motivation for this consistency regularization term.** Specifically, given any pairwise instance $(x_i,x'_i)$ we review the definition of its confidence difference $c(x_i,x'_i)$ below:
> $$c(x_i,x'_i)=p(y'_i=+1|x'_i)-p(y_i=+1|x_i)\in[-1,1],$$
> where $p(y=+1|x)$ denotes the posterior of x belonging to the positive class. Referring to this definition, if $|c(x, x')|>0.5$, $x$ and x' must belong to different classes; and if $|c(x,x')|\le0.5$, x and x' can belong to the same class or different classes, as the posterior difference is insufficient to surpass the classification threshold.
>
> Accordingly, we consider that the pairwise instances whose confidence difference are greater than 0.5 contain more supervised signals, but the other ones may result in noisy signals in the existing ConfDiff method. To explain this, we rearrange the risk of ConfDiff by expanding a prevalent generic form of various loss functions:
> $$R\_{\mathrm{CD}}(g)=\frac{1}{2}\mathbb{E}\_{p(x,x')}\Bigl[\big(\frac{1}{2}-c(x,x')\big)\ell\bigl(g(x),+1\bigr)+\big(\frac{1}{2}+c(x,x')\big)\ell\bigl(g(x'),+1\bigr)\Bigr]+\frac{1}{2}\mathbb{E}\_{p(x,x')}\Bigl[\big(\frac{1}{2}+c(x,x')\big)\ell\bigl(g(x),-1\bigr)+\big(\frac{1}{2}-c(x,x')\big)\ell\bigl(g(x'),-1\bigr)\Bigr] +\frac{1}{2}\mathbb{E}\_{p(x,x')}\Bigl[(1-2\pi)\bigl(g(x)+g(x')\bigr)\Bigr].$$
> where the first and second terms denote the contrastive losses for positive and negative class predictions, respectively; and the third term serves as a regularization. In the first term, the weights $\frac{1}{2}-c(x,x')$ and $\frac{1}{2}+c(x,x')$ determine the contributions of x and x' to the positive class prediction loss, summing to 1 and making $\frac{1}{2}$ as a boundary for prediction directions. These weights lie on opposite sides of the boundary, encouraging one instance to predict more strongly as the positive class, while the other is encouraged to weaken its positive class tendency (i.e., predict as the negative class). In other words, the first loss term ensures x and x' to adjust their predictions in opposite directions, thereby emphasizing the predictive divergence of pairwise instances in the positive class predictions.
>
> This prediction trend holds correctly for pairwise instances with $|c(x,x')|>0.5$. However, when $|c(x,x')|\le0.5$, x and x' may belong to the same class or different classes. In such cases, the prediction trend may lead to samples from the same class being predicted as belonging to different classes, introducing erroneous supervisory signals. Similarly, the second loss term forces to diverge in their predictions for the negative class, which also introduces incorrect supervision for them.
>
> To address the challenge posed by $|c(x,x')|$ being close to 0, where the guidance information becomes imprecise and prediction becomes difficult, we propose setting a threshold $\theta$ to partition the dataset into two subsets: one with relatively precise predictive information (denoted as $D^S$) and the other with comparatively imprecise predictive information (denoted as $D^C$). For $D^C$, we aim to provide additional information to guide the predictions of pairwise instances toward the correct direction. **Specifically, for pairwise instances with small confidence differences, we encourage the model to produce more similar outputs for these pairs. To achieve this, we introduce a consistency regularization term that encourages alignment between the confidence difference and the model’s outputs.** Meanwhile, for $D^S$, we retain the original strategy to preserve the accuracy of predictions driven by this strong guidance.

---

> ### Author Response · Authors · 2024-11-28
>
> Dear respected reviewer,
>
> Thanks again for your valuable review comments that helped improve the quality of our draft significantly.
>
> Please let us know if our answers resolved your questions/concerns.
>
> Many thanks!

---

### Official Review · Reviewer_u74G · 2024-11-01

**Soundness:** 3
**Presentation:** 2
**Contribution:** 2
**Rating:** 5
**Confidence:** 4

**Summary:**

This paper considers weakly-supervised binary classification from noisy confidence difference annotations. It builds on the existing work of risk-estimation based approach and further analyze and modify the terms in the risk estimator. For the proposed risk estimator, error bound is derived and thorough experiments are conducted.

**Strengths:**

- The paper is overall well written. Phrases are generally easy to understand.
- The mathmetical prensentation is sound, and definitions are clear and easy to follow.
- Experiments are thoroughly conducted, with implementation codes attached.
- The background of the problem setting and related methods are well summarized. Authors show a good level of understanding of the problem.
- The problem of tackling noise in the confdiff weakly supervised settings is of high practical importance, and also a theoretically interesting problem.

**Weaknesses:**

- The paper lacks a clear story of presentation of the motivation. There seems exist multiple concepts of "noise". At least there are two different noises: the noise of generated conf diff resulting inaccurate $c$, or the noise of model learned by $|c|<0.5$. They are not clearly described, thus resulting weak motivation for the proposed method.
  - Phrases such as "distribution influenced by real-world noise", "perturb the confidence difference distribution to better fit real-world scenarios", "we observe that the confidence difference values utilized for training in ConfDiff classification are frequently noisy, rather than exact difference between the posterior probabilities of two samples." clearly indicate the motivation is about the noise of the generated conf diffs.
  - At the same time, phrases such as "result in one of the data points being estimated in the opposite direction, introducing noise.
This leads to inaccurate predictive directions even in the absence of noise." seems to talk about another level of noise, noise of the model learned by $|c|<0.5$.
- The important part of consistency regularization term takes a too small part of the whole paper. The motivation and intuition of the modification and how the modification behaves if not clearly demonstrated.
- The other part of contribution, consistency risk of $D^S$ is not well motivated, thus seems irrelevant to the story.
- For Eq. 5, "general form of many commonly used losses" is suddenly introduced without further explanation, such as what specific loss functions belong to this form, or does not belong to this form.
- Too much space is spent on background methods, such as Section 2.

**Questions:**

- For Figure 1
  - What does proportion (x-axis) being more than 1.0 mean?
  - How to control the proportion with a fixed $\pi$? I assume the generation process is that two data points are first i.i.d. generated and then the conf diff is consequently calculated. Does changing the proportion mean that there is a selection mechanism exists during the data generation process, such resulting a skewed data distribution?

---

> ### Author Response · Authors · 2024-11-22
> **Response for Reviewer u74G (Q1-Q4)**
>
> **Q1. The paper lacks a clear presentation of the motivation. There seems exist multiple concepts of "noise". At least there are two different noises which are not clearly described.**
>
> Thank you very much for your valuable review. First, we provide a detailed discussion of the motivation in the "**General Response: Q1. About the motivation of this paper**". Second, the paper indeed presents two descriptions of noise: one in Section 3.1 and the other in Section 4.2. We now provide a detailed explanation and distinction of them.
>
> The noise mentioned in Section 3.1 arises from inaccurate predictive direction information within pairwise instances with $|c(\mathbf{x}, \mathbf{x'})| \leq 0.5 $. For example, consider an instance $\mathbf{x}$ with a posterior probability of 0.2 and another instance $\mathbf{x}'$ with a posterior probability of 0.4, yielding $c(\mathbf{x}, \mathbf{x'}) = 0.2$. According to the discussion in "**General Response: Q1. About the motivation of this paper**", the weights of $\mathbf{x}$ and $\mathbf{x'}$ for the positive class prediction loss are 0.3 and 0.7, respectively, while their weights for the negative class prediction loss are 0.7 and 0.3. This ensures that $\mathbf{x}$ and $\mathbf{x'}$ adjust their predictions in opposite direction, implying $\mathbf{x}'$ is more likely to be classified as positive, which contradicts the actual scenario and thereby introduces inherent noise within $R_{CD}$.
>
> The other type of noise, discussed in Section 4.2, refers to the artificially controllable noise we designed introduced in the confidence difference classification. Specifically, Gaussian white noise is added to the posterior probabilities generated by the probabilistic classifier, thereby indirectly injecting noise into the confidence difference.
>
> &nbsp;
>
> **Q2. The consistency regularization term akes a too small part of the whole paper, with unclear motivation, intuition, and behavior of the proposed modification.**
>
> Thank you for your suggestions. Overall, since the ConfDiff method introduces inaccurate predictive information when $|c(\mathbf{x}, \mathbf{x'})| \leq 0.5 $, we aim to provide precise guidance to encourage the predictions of pairwise instances toward the correct direction. Building on the insight that the consistency regularization term fosters the alignment between confidence differences and model outputs, our method leverages it to encourage more similar outputs for pairwise instances with small confidence differences and reduces the interference of inaccurate guidance on the prediction direction of pairwise instances.
> Please refer to "**General Response: Q1. About the motivation of this paper**" for a more detailed analysis and explanation.
>
> &nbsp;
>
> **Q3. The other contribution of consistency risk of $ D^S $ is not well motivated.**
>
> Thank you for your comment. Please refer to "**General Response: Q1. About the motivation of this paper**".
>
> &nbsp;
>
> **Q4. Eq.5 introduces the "general form of many commonly used losses" without clarifying which specific loss functions fall under this form or are excluded.**
>
> Many thanks for your valuable comments. This function class includes several commonly used loss functions, such as those derived from Generalized Linear Models (GLMs), including the mean squared error (MSE) for linear regression, the logistic loss for logistic regression, the Poisson loss for Poisson regression, the exponential loss for exponential regression, and the cross-entropy loss for neural networks [1].
>
> [1] Linjun Zhang, Zhun Deng, Kenji Kawaguchi, Amirata Ghorbani, and James Zou. How does mixup help with robustness and generalization? In International Conference on Learning Representations, 2021.

---

> > ### Author Response · Authors · 2024-11-22
> > **Response for Reviewer u74G (Q5)**
> >
> > **Q5. About Figure 1.**
> >
> > Thank you for your comments.
> >
> > **What does proportion (x-axis) being more than 1.0 mean?**
> > The x-axis values multiplied by $\pi$ represent the proportion of pairwise instances $(\mathbf{x}, \mathbf{x}')$ with confidence differences $c(\mathbf{x}, \mathbf{x}') \in [-1, -0.5) \cup (0.5, 1]$ relative to all pairwise instances. Thus, the x-axis values effectively serve as a scaling factor used for computing the proportion.
> >
> > **How to control the proportion with a fixed $\pi$?**
> > We do not employ any selection mechanism during the data generation process, thus avoiding any skew in the data distribution. To control the proportions, we first input the data into the classifier to generate their posterior probabilities. Then, we select two instance sets for validation experiments based on the prior $\pi$ from the correctly classified instances. Importantly, the same two instance sets are used across experiments under the same $\pi$ to control for confounding variables. Subsequently, Within each set, instances are sorted by their posterior probabilities, and specified proportions of instances are swapped between the front and rear of the subset to effectively control the proportion of pairwise instances with $\left|c(\mathbf{x}, \mathbf{x}')\right| > 0.5$. This mechanism neither alters the data distribution nor introduces irrelevant variables. Furthermore, we observe a positive correlation between classification accuracy and the proportion, indicating that the pairwise instances with $\left|c(\mathbf{x}, \mathbf{x}')\right| > 0.5$ provide dominant contributions to $R_{CD}$ through precise predictive information.

---

> > > ### Comment · Reviewer_u74G · 2024-11-22
> > >
> > > I thank author for their detailed response. My concerns remain as follows and I will keep my rating.
> > >
> > > - As also mentioned by another reviewer, there are multiple notations of "noise"
> > >   - 1. Inaccurate estimation of $c$.
> > >   - 2. Inaccurate prediction of models learned under $|c| < 0.5$.
> > >   - 3. The noise mentioned in Section 4.2
> > >   - As phrases listed in my initial review, the 1st type of noise seems to be the main motivation of the paper, but authors focus on the 2nd type of noise in rebuttal, resulting inconsistency of motivation.
> > > - For Figure 1, could you explain the process how to plot a line, say the red line of prior = 0.5 for different values of $x$?

---

> ### Author Response · Authors · 2024-11-26
> **Response for Reviewer u74G**
>
> Many thanks for your valuable discussion and response.
>
> **Q1. There are multiple notations of "noise".**
>
> Thank you for your comment. We have provided further clarification on these two issues.. First, we need to clarify that the first type of noise is actually related to the implementation details of estimating $c$ at the experimental level, rather than being the main motivation of this paper. At both the methodological and experimental levels, $c$ is calculated as the difference in posterior probabilities between two instances. The distinction lies in that, at the methodological level, we assume the posterior probabilities used in the analysis are accurate, without involving any estimation. In contrast, at the experimental level, we propose a posterior probability construction method to approximate it. Specifically, we find that the ConfDiff method uses a probabilistic classifier trained on labeled data to generate class posterior probabilities. While this generation method facilitates comprehensive experimental analysis, challenges arise when fitting large-scale real-world datasets and employing sufficiently complex backbones. This often results in the concentration of the posterior probability distribution around certain values, thereby failing to reflect the true distribution influenced by manual annotations in real-world scenarios. Inspired by this, we incorporated a posterior probability construction method based on outlier detection into the probabilistic classifier and calculated confidence differences according to its definition to achieve a more uniform and realistic distribution. The detailed implementation can be found in **Section 4.2** in our paper.
>
> The second type of noise is indeed the main motivation of our paper, and a detailed analysis can be found in the description within "**General Response: Q1. About the motivation of this paper**."
>
> The third type of noise accounts for the fact that in the real world, annotator biases or varying knowledge backgrounds may lead to annotation outcomes that are not entirely precise. To further verify the effectiveness of our method under noisy conditions, we propose adding Gaussian white noise to the posterior probabilities to indirectly introduce noise into the confidence difference. The experimental results strongly demonstrate that our method remains highly competitive even in the presence of noise interference.
>
> Moreover, we acknowledge that certain expressions in the original paper may have been imprecise, potentially leading to confusion regarding the different types of noise and their corresponding motivations. We sincerely thank you for your valuable feedback, which has provided significant guidance for improving our paper. Accordingly, we have revised some descriptions in the manuscript and submitted a new version, which we hope will address your concerns.
>
> &nbsp;
>
> **Q2. For Fig.1, could you explain how to plot a line for different values of $x$.**
>
> Thank you for your comment. The construction of this figure is based on an observation that, when attempting to fit the posterior probabilities of a dataset using outputs from sufficiently complex backbones, the output values tend to yield either very high or very low, resulting in instances being classified with high confidence as either positive or negative. In other words, the posterior probabilities are more concentrated near ${0,1}$, disregarding the more uniform distribution typically observed in real-world scenarios. Therefore, when $\pi=0.5$, given two datasets $ \mathcal{D}\_{1} = \{\bigl(\mathbf{x}\_i, p(y\_i=+1|\mathbf{x}\_i)\bigr)\}\_{i=1}^n$ and $\mathcal{D}\_{2}=\{\bigl(\mathbf{x}'\_i,p(y'\_i=+1|\mathbf{x}'\_i)\bigr)\}\_{i=1}^n$, we can adjust the ordering of data within $ \mathcal{D}\_{1}$ and $ \mathcal{D}\_{2}$ to vary the proportion of pairwise instances with $|c| > 0.5$ over the range $[0,1]$. Specifically, a proportion of 0 corresponds to $ \mathcal{D}\_{1}$ and $ \mathcal{D}\_{2}$ being ordered identically, while a proportion of 1 (equal to $2*\min(\pi,1-\pi)$) corresponds to $ \mathcal{D}\_{1}$ and $ \mathcal{D}\_{2}$ being ordered in strictly reverse sequences.
>
> Based on this observation, we randomly select two datasets, $ \mathcal{D}\_{1}$ and $ \mathcal{D}\_{2}$, both satisfying $\pi=0.5$, and sort their data in ascending order according to posterior probabilities. (After sorting under the aforementioned observation, the first 50% of posterior probabilities are concentrated near 0, while the latter 50% are concentrated near 1.) For example, when the x-axis value is 1, we obtain a proportion of 0.5 for pairwise instances with $|c|>0.5$. To achieve this, we exchange the first 25% and last 25% of instances in $ \mathcal{D}_{2}$. Then, following the definition $\mathcal{D}\_{\mathrm{CD}}=\{\bigl((\mathbf{x}\_i,\mathbf{x}'\_i),c\_i\bigr)\}\_{i=1}^n,\:c\_i=p(y'\_i=+1|\mathbf{x}'\_i)-p(y_i=+1|\mathbf{x}_i)$, we generate a confidence difference classification dataset for training.

---

> > ### Comment · Reviewer_u74G · 2024-11-27
> >
> > Thank you for you detailed reply.
> >
> > For Q1, my concern is resolved that the paper focuses on improving noisy prediction of models learned under $|c|<0.5$. I am also glad to see some sentences in the manuscript are modified to clarify this point.
> >
> > For Q2, I suppose sorting the data make it not i.i.d. and thus violating the data generation assumption stated in the subsection "Formulation of confidence-difference (ConfDiff) classification". I hope authors can further clarify this point in a further revision of the manuscript.
> >
> > Through the discussion, I decide to update my rating to borderline.

---

> > > ### Author Response · Authors · 2024-11-28
> > >
> > > We express our sincere gratitude for upgrading the rating of our paper. The detailed comments and suggestions you provided have undeniably made our paper stronger and more robust. We are truly thankful for the guidance. We will clarify this point in future revisions.

---

### Official Review · Reviewer_u1NV · 2024-11-03

**Soundness:** 3
**Presentation:** 2
**Contribution:** 2
**Rating:** 5
**Confidence:** 3

**Summary:**

The paper studies a special type of weakly-supervised learning known as confidence difference learning. This method leverages confidence differences between unlabeled data pairs to improve classifier training under noisy real-world conditions. By incorporating a noise generation technique and a risk estimation framework that includes consistency risk and regularization, ConfDiff classification demonstrates enhanced robustness and outperforms traditional methods in experiments on benchmark and UCI datasets. Theoretical analyses providing error bounds for the risk estimations further support the method's effectiveness.

**Strengths:**

1. The main theoretical contribution of this paper improved over [1] seems to be the incorporation of consistency regularization and its effect in error bound; based on my understanding, $C_{g}$  bounded the differences between $x$ and $x'$, so that under authors' setup, if the prediction of these data points is close enough, then the perceived generalization error should decrease, which is sensible.

2. After a very coarse examination, the proof of this paper seems to be correct.

3. Encourage instances with smaller confidence differences to produce similar outputs that seem intuitive and sensible, both theoretically and empirically.

[1] Binary classification with confidence difference, NeurIPS 2023.

**Weaknesses:**

1. It appears that the ConfDiff learning requires the classifier to be perfectly calibrated, which is usually infeasible in reality. I think it would be useful to incoperate the results on how different level of calibration error will influence the performance of ConfDiff learning.

2. The implications of the theoritical results in this paper are not discussed, the authors simply present the bound as is, but fail to elaborate any new insights or messages we can draw from the error bounds - what are the dominant terms in the error bound? How is this related to the noisy supervision signal? What properties of your proposed method is related to this error bound? How this bound improves upon the existing bounds? Given the current form of the paper, the audience can only have a vague guess of the above questions.

3. This paper appears to be unclear in many details, why negative risk can lead to overfitting?

4. Suppse we are given a supervision signal $\tilde{c}(x,x') < 0.5$ (smaller confidence difference), then by fitting this objective, wouldn't the $R_{CD}$ inherently encourages $x,x'$ to be similar? Hence making $R_{CRCR}$ trivial?

Minor issues:

1. From line 210-212, shouldn't the $c(x,x')$ be $\tilde{c}(x,x')$ instead?

**Questions:**

How the noise is being defined and formulated in this paper is still a bit puzzling to me, I recommend the authors to improve the writing of the Section 3.1, and maybe considering adding some concrete examples ton make their points.

---

> ### Author Response · Authors · 2024-11-22
> **Response for Reviewer u1NV (Q1-Q5)**
>
> **Q1. ConfDiff learning relies on perfect classifier calibration, which is challenging in practice; studying the impact of calibration errors on its performance would be more meaningful.**
>
> Thank you for your suggestions. In fact, we discarded the data incorrectly classified by the classifier when reconstructing the confidence differences, and this spirit is detailedly implemented in lines 120-144 of the 'main\_strong.py'. Moreover, although the classifier's accuracy during the pretraining phase does not meet the perfect calibration standard, it remains sufficiently high. Therefore, the operation of discarding misclassified data does not significantly affect the scale of the dataset. Detailed information about the accuracy of classifier and the number of misclassified instances is presented in the table below.
>
> | Method      | MNIST  | K-MNIST | FASHION | CIFAR-10 |
> |-------------|--------|---------|---------|----------|
> | **Accuracy**| 0.987  | 0.940   | 0.978   | 0.895    |
> | **Count**   | 14,982 | 14,984  | 14,949  | 9,719    |
>
> &nbsp;
>
> **Q2. The paper presents error bounds without discussing their implications or providing new insights.**
>
> Thank you for your comment. Please refer to "**General Response: Q2. About the analysis for Theorem 1**".
>
> &nbsp;
>
> **Q3. This paper appears to be unclear in many details, why negative risk can lead to overfitting?**
>
> Thank you for your suggestions. Risk is typically non-negative, reflecting the deviation between model predictions and ground-truth values. The objective of optimization is to minimize risk; if risk could be negative, the model could be inclined to find an optimization direction that continually reduces risk on the training data, leading to overfitting by learning noise and performing poorly on test data.
>
> Additionally, [1] highlights that negative empirical risk may be a potential cause of overfitting and experimentally demonstrates a strong co-occurrence of negative risk and overfitting across various models and datasets.
>
> [1] Nan Lu, Tianyi Zhang, Gang Niu, and Masashi Sugiyama. Mitigating overfitting in supervised classification from two unlabeled datasets: A consistent risk correction approach. In International Conference on Artificial Intelligence and Statistics, pp. 1115–1125. PMLR, 2020.
>
> &nbsp;
>
> **Q4. Suppose we are given a supervision signal $ \tilde{c}(\textbf{x}, \textbf{x}') < 0.5 $ (smaller confidence difference), then by fitting this objective, wouldn’t the $ R_{CD} $ inherently encourages $ \textbf{x}, \textbf{x}' $ to be similar? Hence making $ R_{CR} $ trivial?**
>
> Many thanks for your comment. The motivation for our method arises from the observation that small confidence differences may lead to imprecise guidance within $R_{CD}$, particularly when the confidence difference equals zero, resulting in a complete lack of predictive guidance. Our experiments in Figure 1 further demonstrate that pairwise instances with larger confidence differences dominate the contribution to $R_{CD}$, while those with smaller confidence differences contribute minimally. To address this issue, we propose a consistency regularization term that encourages $\mathbf{x}$ and $\mathbf{x'}$ to produce more similar outputs in the model when $|c(\mathbf{x}, \mathbf{x'})|$ is small. A more detailed explanation and analysis can be found in the "**General Response: Q1. About the motivation of this paper**".
>
> &nbsp;
>
> **Q5. From line 210-212, shouldn't the $ c(\mathbf{x}, \mathbf{x'}) < 0.5 $ be $ \tilde{c}(\mathbf{x}, \mathbf{x'}) < 0.5 $ instead?**
>
> Thank you for your correction. We will correct it in future versions.

---

> > ### Comment · Reviewer_u1NV · 2024-11-24
> >
> > Thanks for the detailed rebuttal, however, following question remains:
> >
> > Q1. I fail to see how this is related to the probability calibration.
> >
> > Q4. I don't think this solves my question; as you mentioned, a small confidence difference in $R_{CD}$ hinders the learning process, but how can consistency regularization help? Won't this cause more pairs to have smaller confidence differences and further hurt the performance? Also, if the pair has a small confidence difference, then there is no need to encourage them to produce similar results because their results are already similar (small confidence difference).

---

> > > ### Author Response · Authors · 2024-11-25
> > > **Response for Reviewer u1NV (Q4)**
> > >
> > > **Q4. How does consistency regularization help if small confidence differences hinder learning in $R_{CD}$? Wouldn't this increase small differences and harm performance? If the pair is already similar, why encourage similar outputs?**
> > >
> > > Thank you for your comments. We will address your questions from three aspects.
> > >
> > > **How can consistency regularization help?** The consistency regularization term used in this paper is formally defined as $\bigl(\frac{1}{ \log\left(\left| c(\mathbf{x},\mathbf{x}')\right| + \varepsilon  \right)  } \bigr)  \cdot  \left \| g(\mathbf{x})-g(\mathbf{x}') \right \|_2$ on $D^{C}$, aiming to encourage the classifier to further reduce output differences when the confidence difference is small. In other words, if the confidence difference between instances $\mathbf{x}$ and $\mathbf{x}’$ is small, $g(\mathbf{x})$ should be similar to $g(\mathbf{x}')$, and vice versa. This strategy helps enhance generalization ability and makes the model more robust to noise and perturbations.
> > >
> > > **Won't this cause more pairs to have smaller confidence differences and further hurt the performance?** The core objective of consistency regularization is to enforce consistency in the classifier's predictions for pairwise instances with small confidence differences by constraining the model output. It is important to clarify that our optimization target is the classifier's output $ g(\cdot)$, not the confidence difference $c$. In our setting, $c$ serves as an attribute used for training, functioning as a form of weak supervision. Our goal is not to optimize $c$, but to use it to guide the classifier toward the desired outputs. Therefore, the concern about "causing more pairs to have smaller confidence differences" does not apply here.
> > >
> > > **If the pair has a small confidence difference, then there is no need to encourage them to produce similar results because their results are already similar (small confidence difference).** To clarify this issue, two concepts need to be distinguished: one is the the definition of the confidence difference and how it is generated in the experimental setup. The confidence difference is defined as $c(\mathbf{x}_i,\mathbf{x}'_i) = p(y'_i=+1|\mathbf{x}'_i) - p(y_i=+1|\mathbf{x}_i)$, where the posterior probability represents the accurate posterior probability. In the experimental setup, confidence differences are generated using the output of a probabilistic classifier as an estimate of the posterior probability, which is then used in the definition to create confidence differences for experiments.
> > >
> > > The second concept concerns the meaning of "**result**" in this context. I think the reviewer's question arises from confusion between the experimental-level output of the probabilistic classifier and the classifier outputs optimized by minimizing the risk function at the methodological level. In the methodological context, since we do not know the ground-truth labels, we only have the confidence difference between the two instances, without knowledge of the specific values of the underlying posterior probabilities. Thus, training these instances in the classifier cannot directly guide them to generate similar outputs; in fact, according to the analysis of $R_{CD}$ in "**General Response: Q1. About the motivation of this paper**," $R_{CD}$ may guide them to predict in opposite directions. In other words, "**the similar results**" refers to the outputs of the probabilistic classifier at the experimental level. At the methodological level, we use consistency regularization to mitigate the limitations of $R_{CD}$, encouraging the classifier to produce similar outputs.

---

> > > ### Author Response · Authors · 2024-11-28
> > >
> > > Dear respected reviewer,
> > >
> > > Thanks again for your valuable review comments that helped improve the quality of our draft significantly.
> > >
> > > Please let us know if our answers resolved your questions/concerns.
> > >
> > > Many thanks!

---

> ### Author Response · Authors · 2024-11-22
> **Response for Reviewer u1NV (Q6-Q7)**
>
> **Q6. How the noise is being defined and formulated in this paper is still a bit puzzling to me.**
>
> Thank you very much for your comments. Our paper presents two descriptions of noise, which we now elaborate on in detail below.
>
> The noise mentioned in Section 3.1 arises from inaccurate predictive direction information within pairwise instances with $|c(\mathbf{x}, \mathbf{x'})| \leq 0.5 $. For example, consider an instance $\mathbf{x}$ with a posterior probability of 0.2 and another instance $\mathbf{x}'$ with a posterior probability of 0.4, yielding $c(\mathbf{x}, \mathbf{x'}) = 0.2$. According to the discussion in "**General Response: Q1. About the motivation of this paper**", the weights of $\mathbf{x}$ and $\mathbf{x'}$ for the positive class prediction loss are 0.3 and 0.7, respectively, while their weights for the negative class prediction loss are 0.7 and 0.3. This ensures that $\mathbf{x}$ and $\mathbf{x'}$ adjust their predictions in opposite direction, implying $\mathbf{x}'$ is more likely to be classified as positive, which contradicts the actual scenario and thereby introduces inherent noise within $R_{CD}$.
>
> The other type of noise, discussed in Section 4.2, refers to the artificially controllable noise we designed introduced in the confidence difference classification. Specifically, Gaussian white noise is added to the posterior probabilities generated by the probabilistic classifier, thereby indirectly injecting noise into the confidence difference.
>
> &nbsp;
>
> **Q7. I recommend the authors to improve the writing of the Section 3.1 and add concrete examples to clarify the points.**
>
> Many thanks for your valuable comment. Section 3.1 primarily outlines our motivations. We provide a more detailed discussion in "**General Response: Q1. About the motivation of this paper**", and add concrete examples to clarify key points. We will update Section 3.1 in future revisions.

---

> ### Author Response · Authors · 2024-11-25
> **Response for Reviewer u1NV (Q1)**
>
> Thank you for your discussion and response. We have provided further clarification on these two issues.
>
> **Q1. I fail to see how this is related to the probability calibration.**
>
> Thank you for your comment. I understand that the calculation method of posterior probabilities in the experiments might have caused some confusion. Perhaps you are concerned that using the output of the probabilistic classifier as the posterior probabilities requires the classifier to be highly calibrated. In fact, this method follows the setup presented in [1]. Additionally, we would like to clarify that the confidence difference mentioned in the methodology section is accurately defined as the difference between the posterior probabilities of two instances, specifically $c(\mathbf{x}_i,\mathbf{x}'_i) = p(y'_i=+1|\mathbf{x}'_i) - p(y_i=+1|\mathbf{x}_i)$. In the experimental section, to facilitate comprehensive analysis, we use the output of the probabilistic classifier as an estimate of the posterior probabilities, which is merely an experimental setting. However, we recognize that this estimation may introduce deviations from the true posterior probability distribution. Therefore, in Section 4.2, we propose a posterior probability reconstruction method that detects outliers in the posterior probability distribution using a Gaussian kernel-based discrete posterior probability density estimation. The non-outlier instances are then scaled to achieve a more realistic and uniform posterior probability distribution.
>
> Moreover, we understand your concern regarding the potential noise introduced in the posterior probability calculations if the classifier is not highly calibrated. We have considered this and excluded misclassified instances during the experiments. To further evaluate the impact of noise on our method, we design a method to introduce varying levels of noise to the posterior probabilities and assessed whether our method can achieve stable and accurate results under such artificially induced noise. The experimental results demonstrate that our method maintains increasingly stable and consistent accuracy as the noise ratio increases, with notable improvements in both accuracy and standard deviation, particularly when the noise ratio reaches 100%. This indicates its ability to achieve more competitive results under noise interference.
>
> [1] Wei Wang, Lei Feng, Y uchen Jiang, Gang Niu, Min-Ling Zhang, and Masashi Sugiyama. Binary classification with confidence difference. Advances in Neural Information Processing Systems, 36, 2024.

---

### Author Response · Authors · 2024-11-22
**General Response**

This section provides a general response to the comments on "**the motivation of this paper**" and "**the analysis for Theorem 1**". We would like to thank all of you for your constructive comments, which are very helpful for improving the paper. We have revised our paper carefully. And the detailed point-by-point responses to the comments are given in the corresponding sections for each reviewer.

---

> ### Author Response · Authors · 2024-11-22
> **General Response: Q1. About the motivation of this paper**
>
> **Q1. About the motivation of this paper**
>
> Overall, our motivation arises from the fact that $R_{CD}$ encourages $x$ and $x'$ to predict opposite classes, with one inclined toward positive and the other toward negative. However, this prediction direction may be inaccurate when $|c(x,x')|\leq0.5$, as the two instances may belong to the same or different classes in this case. To address this issue, we propose a ConfDiff classification method based on consistency regularization to mitigate the impact of inaccurate predictions when confidence differences are small.  To describe our work explicitly, we review the research motivation in detail.
>
> First, given any pairwise instance $(x_i,x'_i)$ we review the definition of its confidence difference $c(x_i,x'_i)$ below:
> $$c(x_i,x'_i)=p(y'_i=+1|x'_i)-p(y_i=+1|x_i)\in[-1,1],$$
> where $p(y=+1|x)$ denotes the posterior of x belonging to the positive class. Referring to this definition, if $|c(x, x')|>0.5$, $x$ and x' must belong to different classes; and if $|c(x,x')|\le0.5$, x and x' can belong to the same class or different classes, as the posterior difference is insufficient to surpass the classification threshold. For example, possible scenarios may occur when $|c(x,x')|=0.3$: $p(y_i=+1|x_i)=0.1$, $p(y'_i=+1|x'_i)=0.4$ (both negative); $p(y_i=+1|x_i)=0.3$, $p(y'_i=+1|x'_i)=0.6$ (different classes); $p(y_i=+1|x_i)=0.7$, $p(y'_i=+1|x'_i)=1$ (both positive).
>
> Accordingly, we consider that the pairwise instances whose confidence difference are greater than 0.5 contain more supervised signals, but the other ones may result in noisy signals in the existing ConfDiff method. To explain this, we rearrange the risk of ConfDiff by expanding a prevalent generic form of various loss functions:
> $$R\_{\mathrm{CD}}(g)=\frac{1}{2}\mathbb{E}\_{p(x,x')}\Bigl[\big(\frac{1}{2}-c(x,x')\big)\ell\bigl(g(x),+1\bigr)+\big(\frac{1}{2}+c(x,x')\big)\ell\bigl(g(x'),+1\bigr)\Bigr]+\frac{1}{2}\mathbb{E}\_{p(x,x')}\Bigl[\big(\frac{1}{2}+c(x,x')\big)\ell\bigl(g(x),-1\bigr)+\big(\frac{1}{2}-c(x,x')\big)\ell\bigl(g(x'),-1\bigr)\Bigr] +\frac{1}{2}\mathbb{E}\_{p(x,x')}\Bigl[(1-2\pi)\bigl(g(x)+g(x')\bigr)\Bigr].$$
> where the first and second terms denote the contrastive losses for positive and negative class predictions, respectively; and the third term serves as a regularization. In the first term, the weights $\frac{1}{2}-c(x,x')$ and $\frac{1}{2}+c(x,x')$ determine the contributions of x and x' to the positive class prediction loss, summing to 1 and making $\frac{1}{2}$ as a boundary for prediction directions. These weights lie on opposite sides of the boundary, encouraging one instance to predict more strongly as the positive class, while the other is encouraged to weaken its positive class tendency (i.e., predict as the negative class). In other words, the first loss term ensures x and x' to adjust their predictions in opposite directions, thereby emphasizing the predictive divergence of pairwise instances in the positive class predictions.
>
> This prediction trend holds correctly for pairwise instances with $|c(x,x')|>0.5$. However, when $|c(x,x')|\le0.5$, x and x' may belong to the same class or different classes. In such cases, the prediction trend may lead to samples from the same class being predicted as belonging to different classes, introducing erroneous supervisory signals. Similarly, the second loss term forces to diverge in their predictions for the negative class, which also introduces incorrect supervision for them.
>
> To further validate this perspective, we conduct experiments on the MNIST and CIFAR-10 by varying the proportion of the pairwise instances with $|c(x,x')|>0.5$. The empirical results (in Fig.1) illustrate the accuracy of the binary classifier under different proportions of the pairwise instances with $|c(x,x')|>0.5$. We observe a positive correlation between classification accuracy and the proportion value. These findings demonstrate that the pairwise instances with $|c(x,x')|>0.5$ provide strong guidance and dominate the contribution to $R_{CD}$.
>
> To address the challenge posed by $|c(x,x')|$ being close to 0, where the guidance information becomes imprecise and prediction becomes difficult, we propose setting a threshold $\theta$ to partition the dataset into two subsets: one with relatively precise predictive information (denoted as $D^S$) and the other with comparatively imprecise predictive information (denoted as $D^C$). For $D^C$, we aim to provide additional information to guide the predictions of pairwise instances toward the correct direction. Specifically, for pairwise instances with small confidence differences, we encourage the model to produce more similar outputs for these pairs. To achieve this, we introduce a consistency regularization term that encourages alignment between the confidence difference and the model’s outputs. Meanwhile, for $D^S$, we retain the original strategy to preserve the accuracy of predictions driven by this strong guidance.

---

> ### Author Response · Authors · 2024-11-22
> **General Response: Q2. About the analysis for Theorem 1**
>
> **Q2. About the analysis for Theorem 1**
>
> Thank you for your suggestions.
>
> **1. How is this related to the noisy supervision signal?**
>
> First, as described in the "**General Response: Q1. About the motivation of this paper**", one of our contributions is the partitioning strategy of the dataset into two subsets based on the threshold $\theta$: one with relatively accurate predictive information (denoted as $D^S$) and the other with relatively inaccurate predictive information (denoted as $D^C$). This suggests that noise signals influence the distribution of $c(x,x')$, which in turn affects the choice of an appropriate $\theta$ to balance the subset partitioning strategy, thereby indirectly impacting the error bound.
>
> Specifically, when the noise ratio is low, the majority of samples are assigned to $D^S$. As the sample size $n_1$ increases, both the Rademacher complexity and the term $\frac{4C_\ell}{n_1}\sqrt{2n\mathrm{In}(2/\delta)}$ decrease, enabling the model to quickly converge when training on low-noise samples.
>
> When the noise ratio is high, a larger proportion of samples are assigned to $D^C$. The consistency regularization term encourages the model to output consistent predictions on samples with small confidence differences, thereby reducing the excessive fluctuations induced by noise. Furthermore, from the perspective of the error bound, we control the number of samples in $D^C$ by setting an appropriate threshold $\theta$, thereby balancing the stability and convergence of the error bound. The logarithmic factor $\left| \frac{1}{\log(\varepsilon)}-\frac{1}{\log(\theta+\varepsilon)}\right|$ also ensures that the model's sensitivity to noise remains within a stable range, preventing noisy supervision signals from excessively affecting the error bound.
>
> **2. What properties of your proposed method is related to this error bound?**
>
> The two main contributions we propose—the dataset partitioning strategy and the consistency regularization strategy—are both directly related to the error bound.
>
> For the dataset partitioning strategy, its fundamental idea is to partition the dataset into two subsets based on the threshold $\theta$: one with relatively accurate predictive information (denoted as $D^S$) and the other with relatively inaccurate predictive information (denoted as $D^C$). Through this partitioning strategy, The number of pairwise instance in $D^C$ and $D^S$ can be adjusted according to the distribution of $c(x,x')$, thereby influencing the convergence rate of the error bound through $n_1$ and $n_2$. Furthermore, this strategy effectively smooth the influence of noise by applying consistency regularization.
>
> For the consistency regularization strategy, it encourages the model to output consistent predictions on $D^C$, thus preventing overfitting to noisy features. This strategy reduces the Rademacher complexity on $D^C$, accelerating the convergence of the complexity term in the error bound. Additionally, the term $\left| \frac{1}{\log(\varepsilon)}-\frac{1}{\log(\theta+\varepsilon)}\right|$ effectively controls the range of the error bound, ensuring that the influence of noise on the error bound remains within a stable range.
>
> **3. About the asymptotic convergence properties and rates of the error bound.**
>
> **The asymptotic convergence properties**. The first and second terms of the error bound involve $\mathfrak{R}\_{n_1}(\mathcal{G})$ and $\mathfrak{R}\_{n_2}(\mathcal{G})$, which represent the Rademacher complexities associated with the sample sizes $n_1$ and $n_2$, respectively. Under standard assumptions, as $n_1,n_2\to\infty$, $\mathfrak{R}\_{n_1}(\mathcal{G})$ and $\mathfrak{R}\_{n_2}(\mathcal{G})$ gradually decrease and converge to zero. The third term, containing $\frac{\sqrt{n}}{n_1}$ and $\frac{\sqrt{n}}{n_2}$, also diminishes as the sample sizes increase. Consequently, as $n\to\infty$, $R(\hat{g}_{\mathrm{CRCR}}) \to R(g^*)$.
>
> **The Convergence Rates.** The convergence rates of the first and second terms are governed by the Rademacher complexities $\mathfrak{R}\_{n_1}(\mathcal{G})$ and $\mathfrak{R}\_{n_2}(\mathcal{G})$, corresponding to rates of $O(1/\sqrt{n_1})$ and $O(1/\sqrt{n_2})$, respectively. By decomposing the third term into $\frac{4C_\ell}{n_1}\sqrt{2n\mathrm{In}(2/\delta)}$ and $\left| \frac{1}{\log(\varepsilon)}-\frac{1}{\log(\theta+\varepsilon)}\right|\frac{4\alpha C_g}{n_2}\sqrt{2n\mathrm{In}(2/\delta)}$, it can be observed that the convergence rates of these components are dominated by $O(\sqrt{n}/{n_1})$ and $O(\sqrt{n}/{n_2})$, respectively. Consequently, the overall convergence rate is characterized by $ O\left(\max(\sqrt{n}/{n_1},\sqrt{n}/{n_2})\right)$. Moreover, the balance between $n_1$ and $n_2$ ensures a faster convergence rate, thereby providing stronger guarantees for the model's performance.

---

### Author Response · Authors · 2024-12-04

Dear Area Chair,

We would like to provide a summary following the authors' rebuttal. We sincerely thank all the reviewers for their thorough evaluations and valuable feedback. We are especially grateful to Reviewer u74G for the opportunity to clarify potential issues and for being willing to **update the rating** from 3 to 5. We also appreciate Reviewers wT35 and R1j9 for recognizing our paper's technical soundness and providing positive ratings **6** and **8** prior to the rebuttal, respectively.

We would like to express our sincere gratitude to Reviewer u1NV for their thoughtful review. However, we note that they provided lower scores. In our rebuttal, we carefully addressed the reviewer's feedback and greatly appreciate the two additional questions raised after receiving our response. We have provided detailed replies to these questions but have yet to receive feedback from the second round of review. Consequently, we have not had the opportunity for further discussion with the reviewer. Nevertheless, we are confident that we have thoroughly addressed their concerns.

Regarding the probability calibration concern raised by Reviewer u1NV, we would like to clarify the following:

This issue arises solely at the experimental setup and is unrelated to our method.

Our experiments follows the setup outlined in [1], and motivated by the probability calibration problem, we introduce varying proportions of noise to the posterior probabilities to simulate the effects of calibration errors.

Furthermore, regarding the concern of consistency regularization, we would like to clarify that it is introduced to address the problem of incorrect prediction directions arising from small confidence differences in $R_{CD}$. Our optimization objective targets the classifier's output rather than the confidence difference.

More details are available in our rebuttal response to the reviewers.

According to the input we provided, most reviewers indicated that their concerns were addressed. We sincerely appreciate the reviewers for recognizing the innovation and potential impact of our work.

Thank you!

Sincerely,
The Authors

[1] Wei Wang, Lei Feng, Y uchen Jiang, Gang Niu, Min-Ling Zhang, and Masashi Sugiyama. Binary classification with confidence difference. Advances in Neural Information Processing Systems, 36, 2024.

---

### Meta-Review · Area_Chair_BuF2 · 2024-12-20

**Metareview:**

This work enhances binary classification in weakly supervised learning by leveraging pairwise confidence differences and proposing a consistency regularization-based risk estimator to mitigate noisy supervision. The paper has received mixed feedback from the reviewers. We acknowledge and appreciate the authors' efforts in thoroughly addressing the reviewers' comments and conducting additional experiments to resolve some concerns. However, after a comprehensive review of the feedback and the authors' responses, several issues remain unresolved. These include the lack of a discussion on the implications of the theoretical results, the absence of an analysis of dominant terms in the error bound, and an unclear narrative in presenting the motivation for the work. Based on these considerations, the paper is not recommended for acceptance at ICLR'25.

**Additional Comments On Reviewer Discussion:**

During the rebuttal period, the opinions of two reviewers remained negative despite the authors' efforts to address the concerns raised.

---

### Decision · Program_Chairs · 2025-01-22

Reject